# Monitoring and characterizing soluble and membrane-bound ectonucleotidases CD73 and CD39

**Said A. Goueli**[ID]*, **Kevin Hsiao**

Department of Cell Signaling, Research and Development, Promega Corp. Madison, WI, United States of America

* Said.goueli@promega.com

## Abstract

The success of immunotherapy treatment in oncology ushered a new modality for treating a wide variety of cancers. However, lack of effect in some patients made it imperative to identify other pathways that are exploited by cancer cells to circumvent immune surveillance, and possibly synergize immune checkpoint treatment in those cases. It has been recently recognized that adenosine levels increase significantly in the tumor microenvironment and that adenosine/adenosine receptors play a powerful role as immunosuppressive and attenuating several effector T cell functions. The two main enzymes responsible for generating adenosine in the microenvironment are the ectonucleotidases CD39 and CD73, the former utilizes both ATP and ADP and produces AMP while the latter utilizes AMP and generates adenosine. Thus, these two enzymes combined are the major source for the bulk of adenosine produced in the microenvironment. They were shown to be validated targets in oncology leading to several clinical trials that include small molecules as well as antibodies, showing positive and encouraging results in the preclinical arena. Towards the development of novel drugs to target these enzymes, we have developed a platform that can be utilized to monitor the activities of both enzymes in vitro (biochemical) as well as in cells (cell based) assays. We have developed very sensitive and homogenous assays that enabled us to monitor the activity of both enzymes and demonstrate selectivity of known inhibitors as well as monoclonal antibodies. This should speed up screening for novel inhibitors that might lead to more effective cancer therapy.

## Introduction

Although immunecheckpoint inhibitors showed tremendous success in the clinic, many patients failed to respond to such treatments. Thus, identifying other pathways exploited by cancer cells to circumvent immune surveillance, novel pharmacological agents that will be synergistic with immune checkpoint inhibitors are needed [1]. The insightful observation that a significant increase in adenosine levels was seen in the cancer microenvironment triggered interest in the role played by this nucleoside in tumor onset and progression [2]. Elevated

**Data Availability Statement:** All relevant data are within the paper.

**Funding:** Promega Corp. provided support in the form of salaries for authors to SG and KH, but did not have any additional role in the study design,

data collection and analysis, decision to publish, or preparation of the manuscript.

**Competing interests:** The authors have declared that no competing interests exist.

adenosine concentrations within neoplastic milieu, and activation of adenosine receptors were shown to have powerful immunosuppressive activity and attenuation of several effector T cell functions. Adenosine accumulation in solid tumors at high concentrations stimulated tumor growth and angiogenesis. It also inhibited cytokine synthesis, functions of T cells, macrophages, and natural killer cells [3]. Overexpression of an ecto-5'-nucleotidase (CD73), the enzyme that dephosphorylates extracellular adenosine monophosphate (AMP) to adenosine, within the cancer microenvironment, was recognized as the leading candidate in the generation of a strong immunosuppressive and pro-angiogenic adenosine halo, that facilitates cancer onset and progression, reviewed in [4,5, 6, and 7]. Besides cancer, CD73 expression has been implicated in many other diseases such as autoimmune diseases, ischemia-reperfusion injuries, arterial calcifications, and atherosclerosis [7].

It was also realized that other ectonucleotidases such as CD39 was highly expressed in human ovarian cancer [8]. This enzyme, in sequential reactions, can increase adenosine concentration, by dephosphorylation of adenosine triphosphate (ATP) and adenosine diphosphate (ADP), leading to AMP formation, and the latter is the substrate for CD 73 generating adenosine [8,9]. High extracellular adenosine increased the recruitment of regulatory T ($T_{reg}$) cells by ovarian cancer and correlated with an increase in mortality by suppression of the host spontaneous immune response. The presence of highly expressed CD39 and CD73 on ovarian cancer produces biologically active adenosine which can be as high as 30–60 times higher than that produced by $T_{reg}$ and augmenting the immunosuppressive effect of adenosine on the host immune system [4]. Furthermore, it was recently reported that high level of CD73 expression was found to correlate with a worse prognosis of ovarian cancer patients [10] and in non-small-cell lung carcinoma [11]. In combination with extracellular adenosine, CD73 increased tumor growth and expression of anti-apoptotic Bcl-2 family proteins in tumor cells in vitro [10]. Down regulation of CD73 and CD39 on glioma cells correlated with good prognosis for patients with malignant glioblastomas highlights the validity of CD73 and CD39 as therapeutic targets [12]. Towards this goal, blocking antibodies against CD39 or CD73 showed dampened adenosine production by ovarian cancer cell lines and restored cytotoxicity of NK cells and stimulated proliferation of CD4+ T cells in co-culture with ovarian cancer cells [13]. The observation that inhibition of CD39 and CD73 using antisense oligonucleotides improved immunity against tumors confirm the notion that both ectonucleotidases are promising drug targets, as they act in concert to convert all adenine nucleotides to adenosine [14].

Since CD73-derived adenosine had a wide-ranging effect on the phenotype of both lymphoid and myeloid-derived cells, shaping both the innate and adaptive arms of antitumor immunity, targeting it has become a very high priority in the small molecule, and immunotherapy strategies. Hay et al [15] demonstrated the efficacy of CD73-selective human monoclonal antibodies (MEDI9447) in reversing adenosine-mediated CD4+ T cell suppression, and the pharmacological blockade of CD73 with MEDI9447 was associated with increased antigen presentation and enhanced lymphocyte activation, resulting in a greater release of proinflammatory Th1 cytokines (IFNγ, IL-1β, and TNFα). Blockade of CD73 with MEDI9447 increased the infiltration of several immune cell populations, such as CD8+ T cells and activated macrophages, into the cancer niche with synergic activity upon its combined administration with anti-PD-1 antibodies, further supporting the potential value of relieving adenosine-mediated immunosuppression. This human antibody inhibited CD73 in noncompetitive manner via its binding to the N-terminal domain of CD73 and inhibited the conversion of both membrane-bound and soluble CD73 from the inactive open conformer to the catalytically active closed state [16]. This antibody does not compete with endogenous nucleotide binding to the active site and thus does not require blocking of multiple substrates at the active site and most importantly it inhibits both soluble and membrane bound CD73 through either mono- or bivalent

engagement. Since it does not have an effect on AMP binding to CD73, this antibody has promising clinical utility due to low cross reactivity with other nucleotides. Furthermore, this makes it ideal for combination therapy with existing therapeutic agents that target complementary immune modulating pathways. In recent reports supporting the role of CD73 in tumor growth using engineered mice lacking CD73 displayed resistance to the onset of neoplasia and metastasis, while those receiving therapy targeting CD73 such as small molecule inhibitors [17] or monoclonal anti CD73 antibodies showed antitumor activity [16]. Finally, it has been reported most recently that blocking antibodies targeting the CD39/CD73 immunosuppressive pathway unleashed the immune responses in combination cancer therapies using immune checkpoint inhibitors and chemotherapies [18].

A recent strategy of co-targeting A2AR antagonism and CD73, via antibody directed therapy that engage Fcγ receptors were reported to show effectiveness and indicates that combinatorial treatment with A2AR antagonist and CD73 may provide a promising approach in the clinic [19]. In fact, preclinical trials using a combination therapy that include check point inhibitors showed the A2AR compound AZD4635 induced anti-tumor immunity alone and in combination with anti-PD-L1 (durvalumab) by reversing adenosine-mediated T cell suppression and boosting antitumor immunity [20]. The ability of AZD4635 to block A2A signaling demonstrated the capacity to reduce tumor growth both when administered alone and in combination with PD-L1 checkpoint inhibitors [20]. In other studies, targeted blockade of CD73 enhanced the therapeutic activity of anti-PD1 and anti-CTLA-4 mAbs and provides a potential therapeutic strategy targeting immune checkpoint inhibition [21]. It was reported recently that co-expression of CD39 and CD103 identifies a unique population of CD8 TILs that can only be found within the tumor microenvironment, and higher frequencies of CD103+CD39 + CD8 TILs in patients with head and neck cancer are associated with better overall survival [22].

It is apparent from the above-mentioned observations that the CD39/CD73-adenosine axis may prove to be very promising pathways in immune-oncology, and in fact, great effort is being made to design and develop chemical inhibitors of adenosine receptor as well as ectonucleotidase enzymatic activity and anti-CD73 mAbs, as potential therapeutic approaches aimed at inducing antitumor immune responses.

Because both CD73 and CD39 play a major role (s) in modulation of the immune system and the tumor microenvironment as well as the tumor itself, we have developed an assay platform that monitors the activity of both of these enzymes in vitro (biochemically) and in cell-based assay formats. We further characterized the selectivity profile of compounds inhibiting these enzymes biochemically and in cell-based formats and speculating on which is the preferred route to address this adenosine generating axes. The assays are very simple, homogeneous and are high throughput ready, which facilitate the discovery of next generations inhibitors for CD73 and CD39.

## Materials and methods

### Chemical, reagents, and assay components

Adenosine 5'-(α, β-methylene) diphosphate (AMP-CP), Millipore Sigma (St. Louis, MO). ARL 67156 Trisodium Salt (6-N,N-Diethyl-D-β,γ-dibromomethyleneATP trisodium salt) and POM 1(Sodium metatungstate) were obtained from Tocris Bioscience, R&D Systems (Minneapolis, MN), RPMI Medium 1640, DMEM (4.5g/L D-Glucose, L-Glutamine, and 11mg/L Sodium Pyruvate), Pen Strep, 0.25% Trypsin are from Gibco Life technologies (Thermo Fisher Scientific, Waltham, MA). FBS (Premium Grade Fetal Bovine Serum (Seradigm, VWR Life Science, Radnor, PA), and Hank's balanced salt solution (HBSS) with or without phenol red

from Lonza (Walkersville, MD). Cell lines MDA-MB-231 (human mammary gland/breast adeno-carcinoma), T-47D (human mammary gland ductal carcinoma), SK-MEL-2 (human malignant melanoma), A375 (human malignant melanoma). SK-OV-3 ((human ovary adenocarcinoma), Farage (human B cell lymphoma), DG-75 (human Burkitt's lymphoma), Jurkat Cloe E6-1 (human acute T cell leukemia), and EMEM were purchased from ATCC (Manassas, VA). Active cN-II (human cytosolic 5'nucleotidase) was from NovoCIB SAS (Lyon, France). Active CD73 enzyme (Recombinant Human 5'-Nucleotidase/CD73 Protein, CF) and active CD39 (His-Tag Human) were from R&D Systems, Minneapolis, MN). ADP-Glo, PKA Kinase Enzyme System, PKCα Kinase Enzyme System, SRC Kinase Enzyme System, PI3K Class I Lipid Kinases—p110α/p85α, and Kinase-Glo® Max were obtained from Promega (Madison, WI), Adenosine 5'-Tri-phosphatase from porcine cerebral cortex (Na+/K+-ATPase), Acetate Kinase, Hexokinase, and D-(+)-Glucose were from Millipore Sigma (St. Louis, MO, and Burlington, MA). Other substrates for kinases were included in kits for each kinase tested (Promega corp., Madison, WI). Total pro-tein kit, Micro Lowry Peterson's Modification was from Millipore Sigma (St. Louis, MO). Anti-bodies: for Anti CD73 antibodies: PA 5–11871, PA 5–29750, PA 5–27336, and 410200 were purchased from Thermo Fisher Scientific (Waltham, MA). Ab54217 (7G2), Ab71322, Ab81720 were purchased from Abcam (Cambridge, MA). MABD122 was from Merck Millipore Corpora-tion (Billerica, MA). And Ab 13160S was from Cell Signaling (Danvers, MA). For anti CD39 anti-bodies we used Ab189258 [A1] and Ab108248 [EPR3678(2)] from Abcam (Cambridge, MA). Detailed description of the antibody clones including immunogen and host are shown in S1 Table. Hank's Balanced Salt Solution (HBSS) Buffer with or without phenol red was from Lonza (Walkersville, MD). Protease Inhibitor Cocktail was from Promega (Madison, WI) and PhosStop as phosphatase inhibitor was purchased from Roche Diagnostics (Indianapolis, IN). Precast Tris-Glycine 4–20% gel was purchased from BIO-RAD Laboratories, Inc. (Hercules, CA).

Reagents for SDS-PAGE include Tris-Glycine Running Buffer: 25mM Trizma base, 192mM glycine free base, with 0.1% SDS. Sample buffer with 85mM DTT: 62.5 mM Tris-HCl, pH 6.8, 2.5% SDS, 0.002% Bromophenol Blue, 85mM DTT (freshly prepared), and 10% glycerol.

For Western Blot, TBS buffer (50mM Tris Buffer, pH 7.5, 150mM NaCl), TBST (50mM Tris Buffer, pH 7.5, 150mM NaCl, and 0.1% Tween 20), and antibody dilution buffer (TBST with 5% BSA) were used. Blot-Qualified BSA from Promega Corporation (Madison, WI). Sec-ondary antibodies: Donkey Anti-Rabbit IgG (H+L), and HRP Conjugate, Anti-Mouse IgG (H+L), HRP Conjugate from Promega Corporation (Madison, WI). Protein Markers: Precision Plus Protein™ Dual Color Standard from BIO-RAD Laboratories, Inc. (Hercules, CA), and MagicMark™ XP Western Standard from Thermo Fisher Scientific (Waltham, MA). HRP sub-strate: ECL Western Blotting Substrate from Promega Corporation (Madison, WI).

AMP-Glo™ Assay Kit (Promega, Madison, WI) contains: AMP-Glo™ Reagent I, AMP-Glo™ Reagent II, Kinase-Glo® One Solution, AMP (10mM), and Ultra-Pure ATP (10mM).

Luminescence Microplate Reader (Luminometer); GloMax® Discover multimode micro-plate luminescent reader, Promega Corporation, Madison, WI, or equivalent can be used. Gel electrophoresis, BIO-RAD Criterion™ cell gel unit was from BIO-RAD Laboratories, Inc. (Her-cules, CA), or equivalent.

Western Transfer Blot, iBlot® and/or iBlot®2 Dry Blotting, Invitrogen (Thermo Fisher Sci-entific, Waltham, MA). iBlot®2 NC Regular Stacks was purchased from Thermo Fisher Scien-tific (Waltham, MA), or equivalent. Western Blot Image Unit, any commercial image unit can be used such as ImageQuant™ LAS4000, GE Healthcare Bio-Science AB (Uppsala, Sweden; Pis-cataway, NJ, USA).

Assay Plates, 96-well Treated solid white and clear bottom, 384 well, and low volume 384-well non-Treated white polystyrene or EIA/RIA plate 96-well Half Area non-Treated

white polystyrene (Corning Costar: 3912, 3903, 3917, 3693, 3572, 3693, and 4512, Corning Incorporated, Corning, NY), or equivalent.

## Assay protocols

**AMP-Glo™ assay general protocol.** The AMP-Glo™ assay can be performed in a single tube, 96-, or 384-well plate format.

**ADP-Glo™ assay general protocol.** Assay format is flexible, i.e., volumes of standards (ADP/ATP) utilizing reactions as well as both reagent volumes can be maintained at ration of 1:1:2 regardless of the absolute volumes used in enzyme reactions. Thus, for 96-well format, we recommend using 25μl: 25μl: 50μl.

**Cells preparation for in vivo AMP-Glo™ assay.** Cells from 75 cm$^2$ flasks were trypsinized and counted. Cells were diluted with full medium, followed by plating in tissue culture grade solid white plate at cell density of 25,000 cell per well or as desired using full medium (10% Serum with or without 1% Pen/Strep) for overnight at 37˚C in 5% $CO_2$ incubator.

## I. Monitoring enzymatic activity of purified CD73 and CD39

1. **Determination of enzyme activity of purified CD73.** In a 25 μl reaction buffer (10mM HEPES, pH 7.4, 2mM $MgCl_2$, 1mM $CaCl_2$, and 0.1mg/ml BSA) containing enzyme and AMP substrate at 1μM to 10μM (96-well format), enzyme activity was performed at 23˚C for specified time period and terminated by the addition of equal volume (25μl) of AMP-- Glo™ Reagent I containing 50 μM (final concentration) of known CD73 inhibitor, such as AMPCP. Reactions were mixed well and incubated for additional 30 min at 23˚C. This was followed by the addition of 50 μl of AMP Detection Solution (10μl of AMP-Glo™ Reagent II per ml of Kinase-Glo® One Solution) and incubated for additional 60 min at 23˚C before reading luminescence.

The assay format is flexible, i.e., volumes of the standards (AMP) or any AMP utilizing reactions as well as both reagents volumes can be maintained at the ratio of 1:1:2 regardless of the absolute volumes used (CD73 reaction: AMP-Glo™ Reagent I: Detection Solution). Thus, for 96-well format, we recommend using 25μl:25μl:50μl or multiples of these volumes.

2. **Determination of enzyme activity of purified cytosolic cN-II.** Activity of purified cN-II (Human cytosolic 5'-Nucleotidase) was determined using 25μl reaction buffer (50mM HEPES, pH 7.0, 100mM KCl, 20mM $MgCl_2$, 5mM DTT, 2.5mM dATP, and 0.1mg/ml BSA) containing enzyme and AMP substrate at 10μM (96-well format). Reaction continued at 37˚C for 60 minutes or for a certain time period and terminated by the addition of equal volume (25μl) of AMP-Glo™ Reagent I. Reactions were mixed well and incubated for additional 30 min at 23˚C. This was followed by the addition of 50 μl of AMP Detection Solution (10μl of AMP-Glo™ Reagent II per ml of Kinase-Glo® One Solution) and incubated for additional 60 min at 23˚C before reading luminescence.

3. **Determination of enzyme activity of purified CD39.** Activity of CD39 was determined using reaction buffer that contains 25mM Tris, pH 7.5, 5mM $CaCl_2$, and 0.1mg/ml BSA), with 1μM to 5μM ATP or ADP as substrate, respectively. When ATP is used as a substrate, an equal volume of Kinase-Glo® One Solution was added to CD39 reactions and incubated for 15 min before reading luminescence using a luminometer. When ADP is used as a substrate for CD39, after enzyme reaction completion, an equal volume of AMP Detection Solution with a known CD39 known inhibitor, such as POM 1 at 50μM final concentration was added. Reactions were mixed well and incubated for 60min before

reading luminescence using luminometer. For enzyme dilution, reaction buffer can be used for dilution of enzymes.

## II. Monitoring the activities of ectoCD73 and ectoCD39 in cell-based format

A. **Adherent Cells preparation and assay protocol.** Cells grown in $75cm^2$ flasks were trypsinized and counted, and were diluted with full medium, and plated to a tissue culture grade solid white or clear bottom plate, at cell density of 25K cells in full medium (10% FBS and 1% Pen/Strep) per well, and kept overnight at 37˚C in a 5% $CO_2$ incubator. Fresh medium was added an hour before use by pipetting out ¾ volume of overnight medium from each well and replaced with a ¾ volume of fresh full growing medium. Cells were washed twice with HBSS by taking out whole medium from each well and replaced with fresh 100μl of HBSS to each well. After withdrawing all HBSS, cells were washed one more time with Hank's Buffer before starting treatment by withdrawing all Hank's Buffer out and 100μl cells were added per well in a 96-well plate per cell line per set, experiments were initiated by adding 50μl of each test compound in HBSS and pre-incubated with cells for 5-10min at 37˚C, and reactions were started by addition of 100μl of 2.5X AMP, ADP, or ATP to a final concentration of 5μM ADP or ATP for CD39, or 10μM AMP for CD73 in HBSS. Cells were constantly mixed on a shaker at 100rpm kept at 37˚C.

For CD73 activity using AMP as substrate, 25μl aliquots were withdrawn from each well at the desired time point and added to assay solid white plate (Corning Costar 3912 or equivalent) followed by the addition of 25μl of AMP-Glo™ Reagent I to each well, mixed well and incubated for 30min at 23˚C.This was followed by the addition of 50μl per well of AMP-Glo™ Detection Solution (1ml of Kinase-Glo™ One Solution mixed with 10μl of AMP-Glo™ Reagent II), see AMP-Glo™ TM#384. This was followed by incubating the mixture for 60min at 23˚C, before reading the plate using a luminometer. For comprehensive coverage of AMP Glo assay, please see Mondal et al [23]

For CD39 activity, since it uses both ADP and ATP, we determined its activity using 5 μM final concentration each, as substrate as described above. In a separate plate or microcentrifuge tube, a 25μl aliquots were withdrawn from each well at desired time point and delivered into an assay solid white plate (Corning Costar 3912 or equivalent).

For ADP as substrate, detection of remaining ADP was carried out by preparing AMP Detection Solution (adding 10μl AMP-Glo™ Reagent II to one-ml of Kinase-Glo® One Solution, gently mixing well by inverse tube couple times). To the 25 μl aliquoted from the medium supernatant that was cooled down to room temperature, 25μl of AMP Detection Solution is added to each well, mixed well and incubated for 60min at 23˚C, the plates were read using luminometer.

For ATP as substrate, detection of remaining ATP, was carried out using 25μl withdrawn from the medium supernatant that was cooled down to room temperature and 25μl of Kinase-Glo® One Solution were added to each well, mixed well and incubated for 10min at 23˚C, then the plate was read using luminometer.

B. **Suspension cells.** Cells were harvested and washed two times with HBSS by centrifugation at 1,000 rpm at room temperature for 8~10min, discarding the supernatant and cell pellets were diluted to a desired cell number per ml before experiments.

1. For plate format: Plate 100μl cells per well in a 96-well plate per cell line per set, start the experiments with adding 50μl of each treatment with compounds in HBSS and pre-incubated with cells for 5-10min at 37˚C, and reactions were started by addition of 100μl of

2.5X AMP, ADP, or ATP to a final concentration of 5μM ADP or ATP for CD39, or 10μM AMP for CD73 in HBSS. Cells were constantly mixed on a shaker at 100rpm kept at 37˚C.

2. For microcentrifuge tube (1.5ml): 250μl cells in HBSS were used per sample per tube, by adding 50μl compound in HBSS, then reactions were started by addition of 200μl of 2.5X AMP, ADP, and ATP to a final concentration of 5μM ADP or ATP for CD39 activity, or 10μM AMP for CD73 activity). Cells were constantly mixed on a shaker at 100rpm keep at 37˚C (no Vortex).

Cell suspensions were centrifuged at 1,000 rpm (plate format) or 2,500 rpm (microcentrifuge tube format) for 2-3min and 25μl samples were withdrawn out of each well or sample tube for detection of reaction products as shown above for adherent cells.

### Electrophoresis and western blotting of cell lysates

**Sample preparation.** Cells were grown in one 75cm$^2$ flask till 80~90% confluence. Cells were treated with trypsin and collected using 300 g for 10min and lysed with 1ml of lysis buffer containing protease and phosphatase inhibitors. The lysis buffer contains 50mM Tris, pH 8.0, 150mM NaCl, 0.5% Triton X-100, 1mM Na$_3$VO$_4$, 1mM DTT, 5mM EGTA, 5mM EDTA, 10mM NaF, 10mM Na$_2$P$_2$O$_7$, with protease inhibitors (80μM Aprotinin, 2mM Leupeptin, 1.5mM Pepstatin A, 104mM AEBSF, 1.4mM E64, 4mM Bestatin) and phosphatase inhibitors (PhosSTOP, one tablet per 10ml of lytic buffer). Cell lysates were sonicated for 8 pulses on ice (Output set at 3.5 for micro tip, and 35% of Duty Cycle SONIFIER 450, Branson (Danbury, CT), and kept on ice after sonication for 5min. Cell lysates were centrifuged in pre-chilled 4˚C centrifuge for 10min at 10,000 g, and the supernatant was tested for total protein estimate using Total Protein kit from Millipore Sigma (see material section) and cell lysate was frozen at -70˚C for electrophoresis where 10μg or 20 μg per sample in sample buffer was heated up at 95˚C for 5 min and run in Tris-Glycine gel followed by dry transfer for Western blot detection following the instruction for using iBlotⓇ and/or iBlotⓇ2 Dry Blotting with BlotⓇ2 NC Regular Stacks.

**Western blot.** After transferring from dry blot, nitrocellulose membranes were quick washed twice with 1X TBS Buffer. Membranes were blocked with 1X TBS containing 5% BSA for 1 hour at room temperature. Membranes were then washed with TBST three time, 10min each wash with constant rocking. Primary antibody with proper dilution in TBST containing 5% BSA were added and incubated overnight at 4˚C. The primary antibodies were discarded followed by three times washes with TBST,10min each, with constant rocking. Secondary antibodies with proper dilution in TBST containing 5% BSA were added and incubated for 1 hour at room temperature. Secondary antibodies were discarded, washed twice with TBST 10-min each, then with TBS washed twice with constant rocking. Membranes were kept in TBS before adding HRP substrate, and development and imaging for documentation.

## Results and discussion

As presented in the introduction, the two ectonucleotidases CD39 and CD73 are validated drug targets. [24, 25, 23,24]. CD39 is an integral membrane protein that hydrolyses ATP and ADP in a calcium and magnesium dependent reaction generating AMP. It is activated upon glycosylation and translocation to the cell surface membrane where it displays its enzyme activity as an ectonucleotidase. CD73, the potent suppressor of antitumor immune responses [5], is a dimer anchored to the plasma membrane via a C-terminal serine residue that is linked to glycosylphosphatidyl inositol (GPI) without any membrane traversing protein. There is also a soluble form of this enzyme that exists through shedding from the membrane by the action

of phosphatidylinositol-specific phospholipase which hydrolyzes the GPI anchor. Interestingly, this enzyme is inhibited by ATP and ADP in a competitive manner to its substrate AMP. In fact, the most potent inhibitor for CD73 is an ADP analog (AMP-CP). There is also cytosolic 5'-nucleotidase that is structurally distinct from the ectonucleotidase CD73.

Because of the significance of these enzymes and the role they play in multiple signaling and inflammatory pathways we developed an assay that can be used to monitor their activities not only in a biochemically purified enzyme forms but also in a cell-based assay. The assay platform is simple, easy to perform, homogenous and less susceptible to generation of false hits and ready for use in high throughput screening (HTS) programs searching for modulators of these enzymes.

The principle of the assay is shown in S1 Fig where ATP and ADP as substrates for CD39 (S1A Fig) can be monitored by two distinct assay formats, decrease in ATP utilized by the enzyme or an increase in ADP formation generated. When ADP is used as substrate for CD39, the remining ADP is converted to ATP and the latter is detected using luciferin/luciferase reaction. The activity of CD73 is monitored by the utilization of AMP substrate (S1B Fig), and thus a decrease in AMP concentration (decrease in luminescence) is proportional to the activity of the enzyme; i.e., a reciprocal relationship between the RLU values generated and the activity of the enzyme.

Since both enzymes are ectonucleotidases, we used an isosmotic medium to keep cellular integrity and we then added the substrates to the medium and monitored changes in the nucleotides concentration in the medium without disrupting cellular membranes. Similar detection steps were used as described above for the purified enzymes.

Fig 1A, 1B and 1C show that we were able to determine the concentrations of AMP, ADP, and ATP at nanomolar concentrations. Thus, we can monitor minute changes in the concentrations of these metabolites in a biochemical reaction as well as in the medium aspirated from cell culture plates.

## Monitoring the enzymatic activity of soluble and membrane-associated CD73

A. **Monitoring enzymatic activity of soluble purified CD73.** Since AMP is the major substrate for CD73 we generated standard curve to establish the linearity of the assay. As shown in the figure, we can detect as low as 30 nM AMP in the reaction. AMP determination is based on the conversion of AMP to ADP followed by converting ADP to ATP followed by detection of ATP using luciferase/luciferin reaction. After establishing the

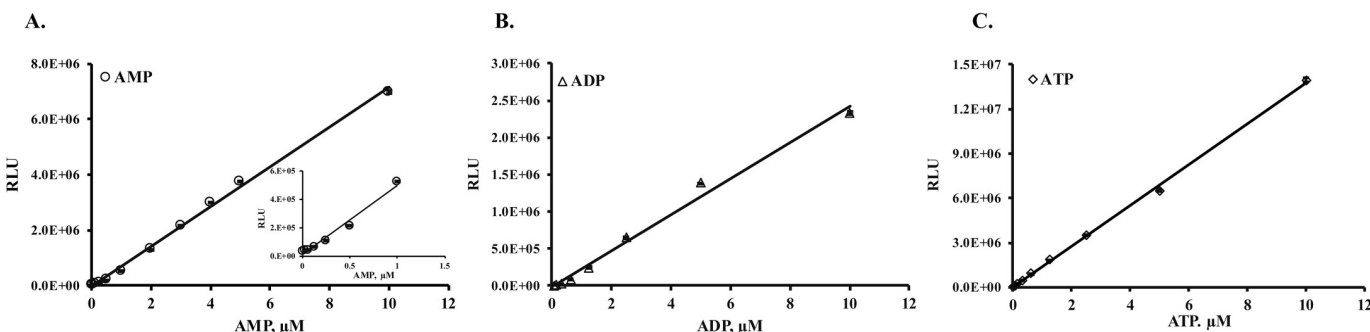

**Fig 1. Assay sensitivity in monitoring AMP, ADP, and ATP concentrations.** (A) Titration of AMP in the presence and absence of 100μM ATP to demonstrate that AMP can be determined in the presence or absence of ATP thus showing no interference from ATP in the reaction since it is removed by adenylate cyclase before conversion of AMP to ATP. The inset is an expanded scale of AMP at lower concentrations to show the sensitivity of the AMP detection. (B) ADP titration, and (C) ATP titration were carried out as mentioned in scheme IA. As shown in the figure, the limit of detection (LOD) reaches 10nM or less. Each point represents an average of a triplicates and results shown are mean ± SD. SD, standard deviation.

linearity of AMP concentrations, we tested purified CD73 for its enzymatic activity by monitoring depletion of AMP, i.e., reciprocal relationship between enzyme activity and luminescence output (Fig 2).

B. **$K_m$ determination for AMP and inhibitor selectivity.** Fig 3A shows that the assay is very sensitive by detecting low concentration of CD73 using AMP as substrate. The amount of enzyme required for 50% of maximum activity was as low as 80 picogram (pg) of CD73 for a 20 minutes reaction, and 200 pg for 5-minute reaction. In order to use the assay for determination of kinetic parameters of the ectonucleotidases, we used the initial velocity values of CD73 enzyme activity at varying concentrations of AMP substrate to calculate Km value for AMP and $IC_{50}$ of known inhibitor. As shown in Fig 3B, CD73 has a $K_m$ of 2.53 μM. Next, we tested the inhibition of enzyme activity using Adenosine 5′-(α, β-methylene) diphosphate (AMP-CP), a potent and selective inhibitor for CD73 using two different concentrations of AMP (5 and 10 μM) in a 5-minute reaction time. The results in Fig 3C show that we obtained an $IC_{50}$ of 0.157 μM and 0.30 μM for the enzyme using 5 and 10 μM AMP as substrate, respectively, indicating a competitive inhibition mode of action for the inhibitor with respect to the substrate AMP, which is similar to what was reported by others, $IC_{50}$ = 0.236 μM [25, 26].

We also tested our assay using purified cytosolic CD73 (cN-II) so it can be tested for inhibition by the known ecto CD73 selective inhibitor AMP-CP. Fig 4A shows an activity curve for cN-II enzymatic activity using 25 μM AMP as substrate and we show that the amount of enzyme that gives 50% of maximum activity was 0.0366 mU attesting to the sensitivity of the assay. We also observed that the presence of dATP in the reaction, which is carried out at its optimal temperature, 37˚C, significantly augmented the activity of the enzyme (results not shown), that is similar to what was reported by others [27].

It is interesting that AMP-CP showed no inhibition of the cytosolic enzyme which is in accord with what has been reported in the literature [28] while ecto CD73 showed again

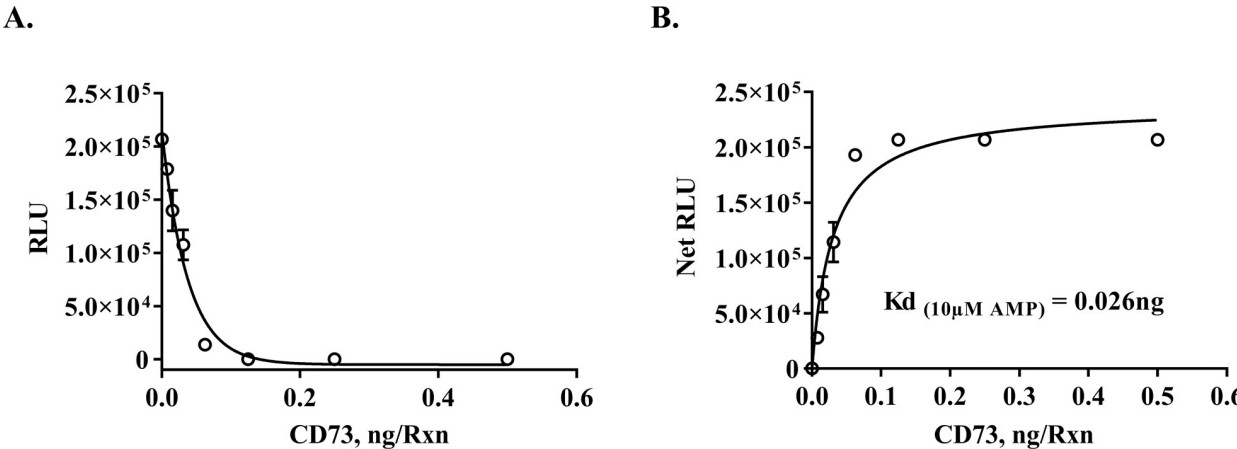

**Fig 2. Determination of enzyme activity of purified CD73.** A. Activity determination of recombinant human 5'-Nucleotidase/CD73 using increasing concentrations of enzyme and 10μM AMP substrate. The reaction was carried out at 23˚C for 30min using AMP-Glo Assay System as described in Materials and Methods section. Activity of CD73 is monitored by how much AMP has been consumed in (A), RLU corresponds to the amount of AMP remaining and thus the activity of the enzyme is reciprocally correlated with RLU (see S1A Fig; (B) Net RLU after subtracting the control (no-enzyme) from the RLU values at each point of enzyme concentration. The experiment was done in triplicates; results shown are mean ± SD. SD, standard deviation.

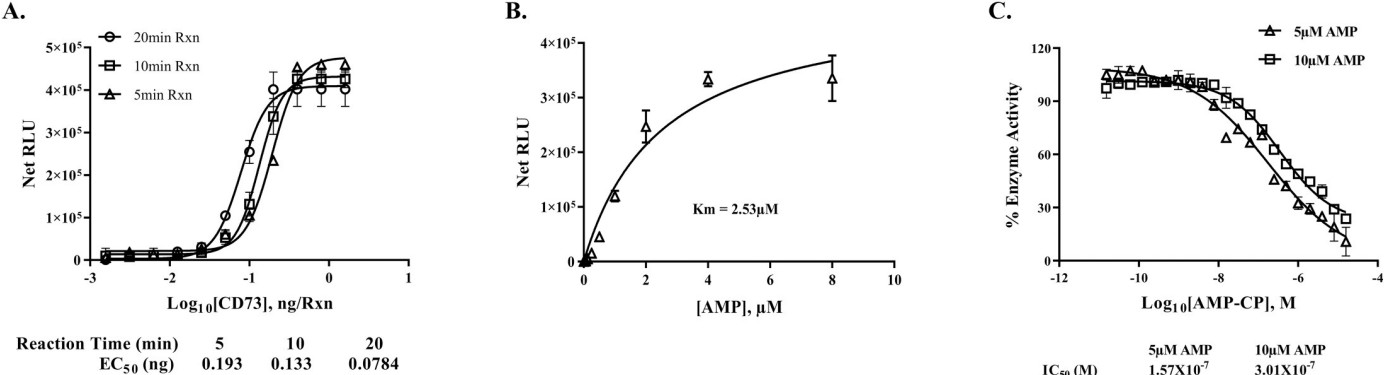

**Fig 3. Effect of enzyme concentration, substrate concentration and IC$_{50}$ determination using purified recombinant human CD73. A)** Time course study using recombinant human 5'-Nucleotidase CD73 protein titration and 25μM AMP for 5,10, and 20 minutes reaction time at 23˚C. Activity was determined using AMP-Glo assay. Data are shown as net RLU after subtracting values for no enzyme control from the RLUs values at each point of enzyme concentration. EC$_{50}$ represents the amount of enzyme required for 50% maximal activity. Each point represents the average of triplicates; error bars represent SD. **B)** Determination of Km value for AMP using 0.01ng recombinant human 5'-Nucleotidase/CD73 protein per reaction and varying AMP concentrations for 5min reaction at 23˚C followed by AMP-Glo assay protocol. Data shown as net RLU vs. AMP concentration. The experiment was done in triplicates; results shown are mean ± SD. **C)** Determining IC50 for AMP-CP using 0.1 ng of recombinant human 5'-Nucleotidase/CD73 protein and either 5μM or 10μM AMP and different concentrations of inhibitor (AMP-CP). Reactions were carried out for 5min at 23˚C and activity was monitored using AMP Glo assay. Experiments were done in triplicates, and results are shown as mean ± SD.

inhibition by the compound with an IC$_{50}$ of 0.3 μM (Fig 4B). Thus, the assay not only can determine soluble CD73 (purified enzyme) and membrane-associated CD73 (see below), but it also confirms the selectivity of the inhibitor AMP-CP towards the ecto CD73 but lacks activity towards structurally non-related cN-II nucleosidase.

After successfully determined CD73 enzymatic activity in its pure form, we then attempted to monitor the activity of membrane-associated ectonucleotidase CD73 of several cell lines. As shown in Fig 5A, it is apparent that the activity of CD73 as measured by the decrease in luminescence upon AMP utilization increases in all cell lines tested with incubation time up to 2 hrs.

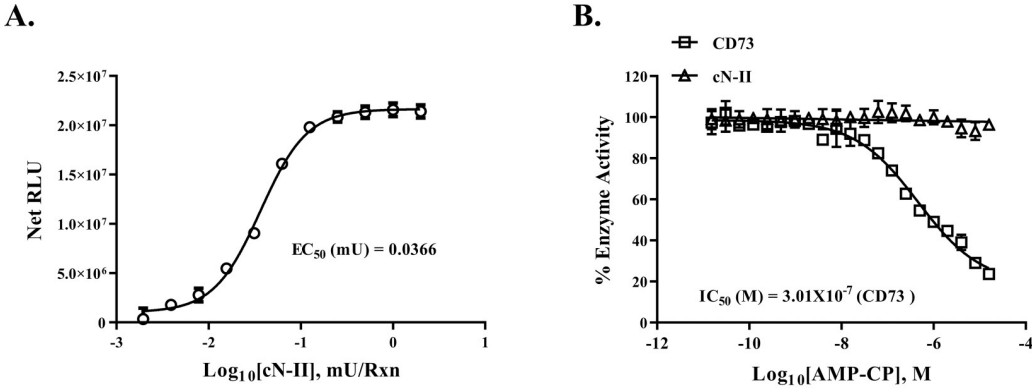

**Fig 4. Determination of enzyme activity of purified cytosolic cN-II and inhibitor selectivity. A)** Enzyme reaction for cN-II (Human cytosolic 5'-Nucleotidase) was carried out using increasing concentrations of enzyme and 10μM AMP for 60min at 37˚C and activity was determined following AMP-Glo assay as described in Materials and Methods section. The results are shown as net RLU after subtraction of no enzyme control. The experiment was done in triplicates; results shown are mean ± SD. **B)** Selectivity of the inhibitor AMP-CP against CD73 and cytosolic cN-II was determined using purified CD73 (0.1 ng/Rx) and cN-II (0.6 mU/Rx) by AMP-PC using 10μM AMP substrate at 23˚C for 5 min (CD73) and at 37˚C for 30 min (cN-II). Activities were determined following AMP-Glo protocol. The inhibitor inhibited CD73 activity with an IC$_{50}$ value of 3x10$^{-7}$M with minimal or no inhibition of cN-II. Experiments were done in triplicates; results shown are mean ± SD. **C)** Monitoring enzymatic activity of membrane-associated CD73 in cell-based assay.

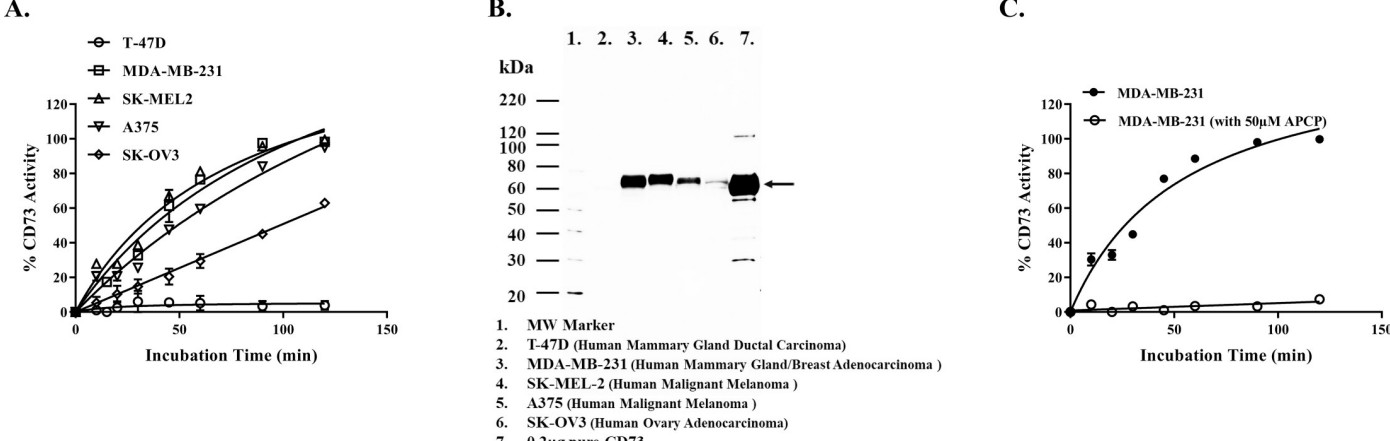

**Fig 5. Determination of membrane bound CD73 and effect of its inhibitor (AMP-CP) on soluble and membrane bound CD73 activity. A)** Five cell lines were evaluated for their CD73 enzyme activity in intact cells using AMP Glo protocol for cells as described in the method section. Time course of enzyme activity with 10 μM AMP following AMP-Glo protocol. Cells: T-47D (circle), MDA-MB-231(square), SK-MEL2 (triangle), A375 (reverse triangle), and SK-OV3 (diamond). **B)** Determination of abundance of CD73 in membranes of five cell lines using western blotting. Cell lysates (10μg) from each cell line and pure CD73 (0.2μg) as positive control (lane 7) were run on gels and immunoblotted using primary antibodies (anti- NT5E/CD73, Cell Signaling), incubated overnight at 4°C followed by HRP-ECL as described in the method section. **C)** Membrane associated CD73 bound MDA-MB-231 activity was determined in presence (open circle) and absence (closed circle) of 50μM AMP-PC. Reactions contained 25k MDA-MB-231 cell per well and incubated at 37°C with 10μM AMP in final 250μl per well. Aliquots of 25μl per sample were withdrawn at each time point and activity was determined following AMP-Glo assay. Each point represents the average of triplicates; the error bars represent the SD.

It is also apparent that MDA-MB-231 has the highest activity and SK-OV3 showed the lowest, while the other cell lines (T-47D, SK-MEL2, and A375) gave intermediate values for the enzyme activity or none. To confirm that the activity of ectonucleotidase CD73 we observed, reflects the level of the membrane associated enzyme, we isolated membranes from these same cells and solubilized them for SDS gel electrophoresis and western blotting using anti CD73 antibodies. As shown in Fig 5B, when 10 μg of cell lysate of each cell line were used, the density of the bands that correspond to CD73 was highest in MDA-MB-231 and lowest in SK-OV3 cells while the other cell lines showed intermediate band densities corresponding to the enzyme activity determined using whole cells. These data support our enzyme activity data and that the assay can monitor the activity of CD73 of purified enzyme as well as membrane-associated enzyme.

It is critical for the assay to be used successfully in monitoring the activity of ecto CD73 on membrane to demonstrate that inhibition of CD73 that we observed with purified soluble enzyme can be also demonstrated using membrane-associated enzyme. To carry out these studies we selected MDA-MB-231 cell line since it contains the highest ecto CD73. As shown in Fig 5C, AMP-CP inhibited the enzyme activity while the activity of the enzyme was not affected with time in the absence of the inhibitor.

To check once more for the selectivity of the assay in monitoring the activity of CD73, we tested purified CD73 using known CD39-inhibitors. As shown in Fig 6A, the two inhibitors ARL 67156, and POM1 had no effect on CD73 enzyme activity while AMP-CP inhibited purified enzyme with $IC_{50}$ 0.588 μM (Fig 6B), and membrane associated enzyme with $IC_{50}$ of 2.1 μM (Fig 6B).

Three compounds with inhibitory activity against CD73 (AMP-PC, solid circle) and against CD39 (ARL 67156 and POM 1, square and triangle, respectively) were tested using purified CD73 and MDA-MB-231 bound CD73. (A) Purified CD73, 0.1ng per reaction with 1μM AMP for 5 min reaction at 23°C, and (B) cell-based MDA-MB-231, 25K cells per well with 5μM AMP for 90min at 37°C. Each point represents the average of triplicates; the error bars represent the SD

**A. Purified CD73**

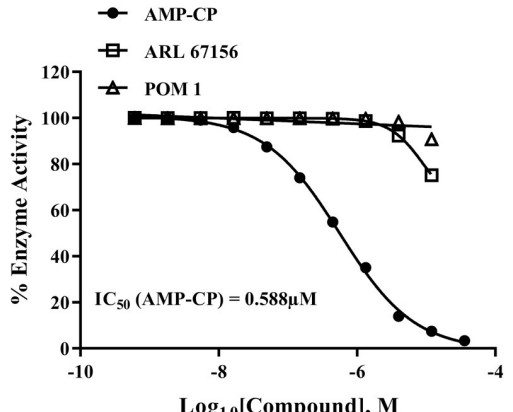

**B. CD73 Cell-Based MDA-MB-231**

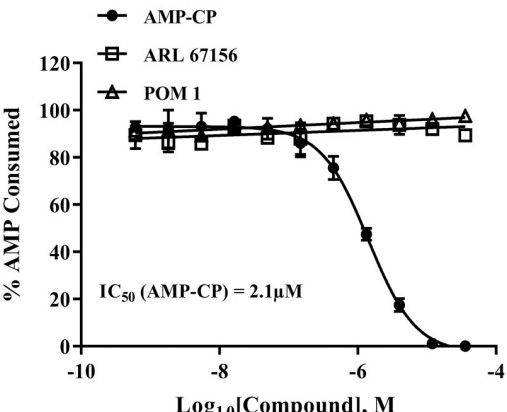

**Fig 6. Selectivity of various inhibitors against soluble CD73 and membrane bound CD73 enzyme using assay protocol for purified and cell-based assays.**

To find out whether CD73-selective antibodies block the activity of membrane-associated CD73, we tested nine CD73 commercially available antibodies, plus a CD39 selective antibodies as control antibody (see S1 Table) for their effect during incubation with MDA-MB-231 cells for up to 16 hrs. to assess the mode of action of these antibodies (Fig 7A and 7B). We even used two antibodies that recognize the same epitope but supplied by two different vendors to check on the quality of data generated using our platform. As expected, the CD39 antibody and the control (no antibodies) showed minimal effect on the activity of the membrane-associated CD73 (maximum of 10% at 10 μg/ml for 6 hrs. of incubation with the cells), while the others showed variable effect on the enzymatic activity with antibodies #4, #5, demonstrating highest potency towards CD73; 75% and 62% inhibition of CD73 at 1 μg/ml and 5 μg/ml for antibody #4 and #5 respectively (Fig 7B).

**C**ommercially available antibodies against CD73 were tested using cell-based membrane-associated CD73 from MDA-MB-231 following AMP-Glo cell-based assay. (A) Using 9 different commercially available antibodies against CD73 with an antibody against CD39 as a negative control. For details on the antibodies tested including immunogen and host, see S1 Table. Antibodies were applied at 10μg/ml per well and incubated with MDA-MB-231 cells for 1, 3, 6, and 16 hrs. Activities are expressed as percentage of control (cells without antibodies). (B) Effect of antibodies concentration of the most effective antibodies (Ab4 and Ab5) and the control CD39 selective antibodies (Ab10) on the activity of membrane bound CD-73. Activities are expressed as percentage of control (cells without antibodies). All antibodies were tested using 25K cells per well cells and incubated at 37˚C. Activities were determined in the presence of 10μM AMP substrate for 30min at 37˚C. Each point represents the average of triplicates; the error bars represent the SD

A. **Monitoring the enzymatic activity of soluble and membrane-associated CD39.** As we discussed in the method section, CD39 can use both ATP and ADP as substrates and therefore we designed two different protocols to assess the activity of this enzyme in soluble purified enzyme form as well as membrane-associated cell-based form. The first protocol monitors the depletion of the substrate while the other monitors the formation of the product ADP. The results in Fig 8 show that the activity of the enzyme can be

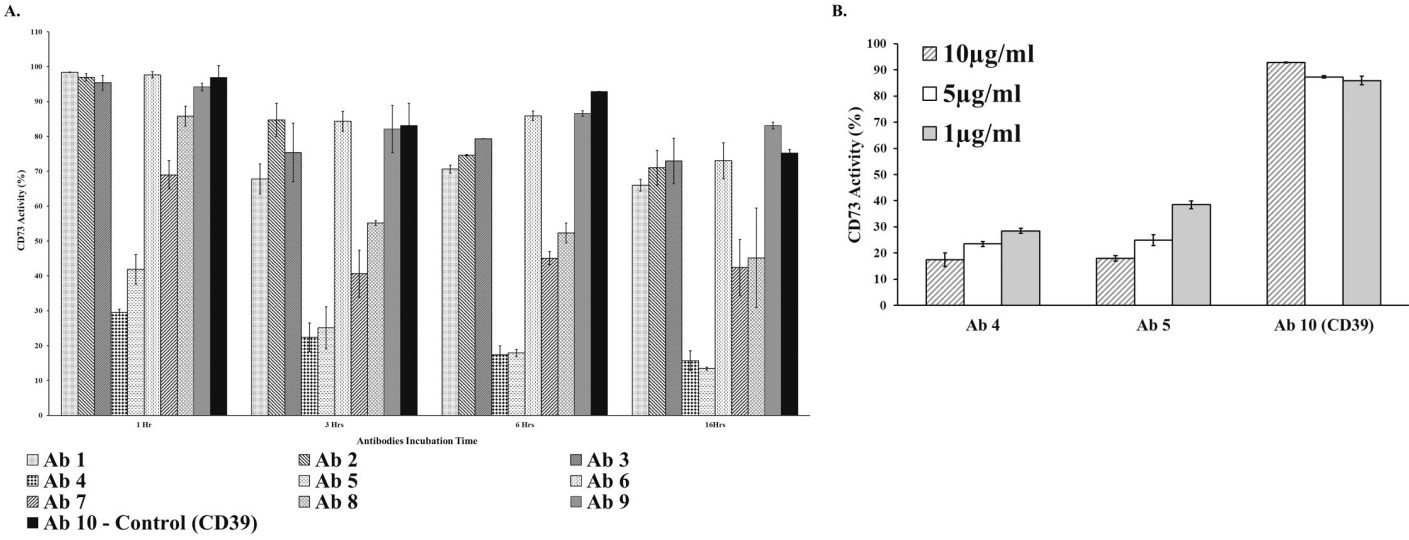

**Fig 7. Effect of various anti-CD73 antibodies on blocking the activity of CD73 bound MDA-MB-231 cells.**

monitored by both protocols with $EC_{50}$ for CD39 of 0.052 ng/reaction using 10 μM ATP as substrate (Fig 8A) and 0.019 ng/reaction using 10 μM ADP as substrate (Fig 8B).

We then followed this by checking the ability of the two protocols in assessing the selectivity of known CD39 inhibitors, such as 6-N, N-Diethyl-D-β-γ-dibromomethylene ATP trisodium salt (ARL67156) and sodium metatungstate $3NaWO_4.9 WO_3.H_2O$ (POM1). The $IC_{50}$ for both inhibitors with ATP (10 μM) as substrate and 0.1 ng enzyme were $4.48 \times 10^{-6}$ M and $0.33 \times 10^{-7}$ M for ARL 67156 and POM1, respectively Fig 8C). The $IC_{50}$ for the same inhibitors using 10 μM of ADP as substrate and 0.1 ng of enzyme were $2.8 \times 10^{-4}$ M and $3.7 \times 10^{-6}$ M, respectively (Fig 8D).

It is apparent that POM1 is more potent inhibitor of soluble CD39 using both assay formats. To check whether we could assay the activity of membrane-associated CD39, we used three cell lines for these experiments. As it is apparent, human B Cell lymphoma (Farag) showed the highest CD39 activity while human Burkitt's lymphoma (DG-75) and human acute T cell leukemias (Jurkat E6-1) showed very low activity (Fig 9A).

To confirm that the activity of the membrane associated enzyme observed with Farag cell truly represents CD39 enzyme activity and not as much with the other cells, we ran a western blotting of cell membranes using anti CD39 antibodies. As shown in Fig 9B, the intensity of the bands corresponding to CD39 is proportional to the enzyme activity observed in Fig 9A. Testing the inhibitor potency on cell based membrane-associated CD39, we incubated Farage cells with 5 μM ATP, in the presence or absence of 50μM POM1. As shown in Fig 9C, CD39 activity was observed in the absence of POM1 while its activity in the presence of POM1 was inhibited.

To check on the selectivity of the inhibitors, we tested the effect of CD73 inhibitor AMP-CP and the CD39 inhibitors ARL67156 and POM1 using 10 μM ATP or 10 μM ADP as substrates using purified CD39 (Fig 10A and 10B). It is apparent that AMP-CP has no inhibitory effect on CD39 but POM1 and ARL67156 inhibited its activity with POM1 showing higher potency as we observed earlier (Fig 9). This selectivity was maintained using cell-based membrane associated CD39 and ATP as well as ADP as substrates (Fig 10C and 10D).

### A. ATP as Substrate

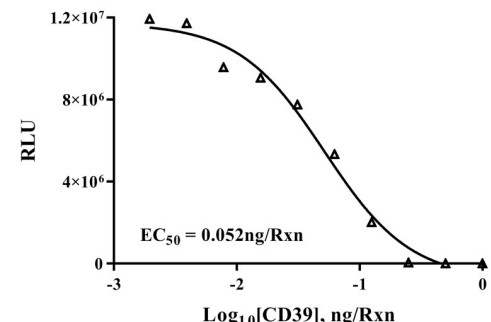

$EC_{50} = 0.052 \text{ng/Rxn}$

### B. ADP as Substrate

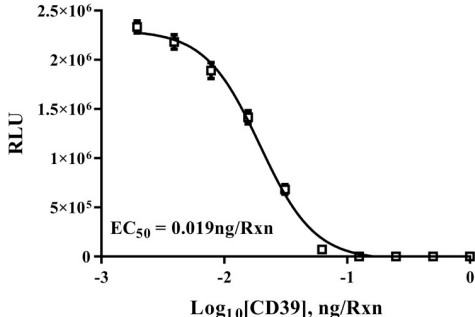

$EC_{50} = 0.019 \text{ng/Rxn}$

### C. ATP as Substrate

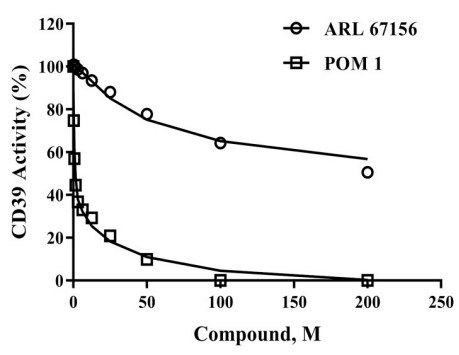

### D. ADP as Substrate

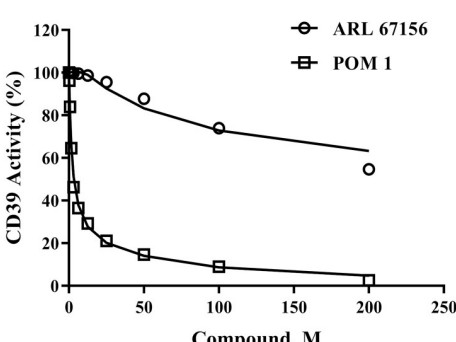

**Fig 8. Determination of the enzyme activity and inhibitory potency on purified CD39.** CD 39 enzyme activity was determined using either ATP (A) or ADP (B) as substrate and various enzyme concentrations as described in the method section. The results show that purified CD39 can use both substrates ATP (8A, 10μM) and ADP (8B, 10μM) at 37°C for 30min reaction, followed by AMP-Glo assay. The experiment was done in triplicates; results shown are mean ± SD. Inhibitors study was carried out using purified CD39 (0.1 ng) with 10 μM ATP (C) or 10 μM ADP (D) as substrates are shown. Two known CD39 inhibitors (ARL 67156 and POM1) were tested for their inhibition of purified CD39 following assay protocol. Results show the percent remaining activity in the presence of ARL 67156 (circle) and POM1 (square) in comparison to no compound control. Both compounds inhibited CD 39 enzyme activity with either substrate but with different potency. Each point represents average of a typical experiment done in triplicates; results shown are mean ± SD.

Finally, since CD39 uses ATP as a substrate, that is similar to kinases, we tested for the selectivity of the most potent CD39 inhibitor POM1 on different kinases to validate whether it is selective for CD39 or it is a promiscuous inhibitor and inhibiting diverse set of kinases. It is apparent from the results presented in Fig 11A that POM1 is not selective at all against four protein kinases (PKA, PKCα, Src, ACKα), sugar kinase (HK), lipid kinase (PI3Kα), and K$^+$/Na$^+$-ATPases. Since its inhibitory potency for many kinases is in the very low micromolar concentrations range, this compound cannot be used as a selective inhibitor for CD39 since many of those enzymes listed here are present either on cell membranes or may leak out of the cells thus distorting the results significantly. We believe that because POM1 is mainly composed of the heavy metal metatungstate, it is not surprising that it inhibited the other enzymes with similar potency to its effect on CD39. In contrast, the selective inhibitor for CD73 AMP-CP showed relatively higher selectivity since it has not inhibited all those enzymes tested except that at very high concentration compared to those used for CD73 where it showed some inhibitory effect (Fig 11B). Therefore, it may be prudent to focus on CD73 with small molecules inhibitors or selective antibodies but with CD39, the focus should be placed on use of selective CD39 antibodies to evade any cross inhibition of other targets.

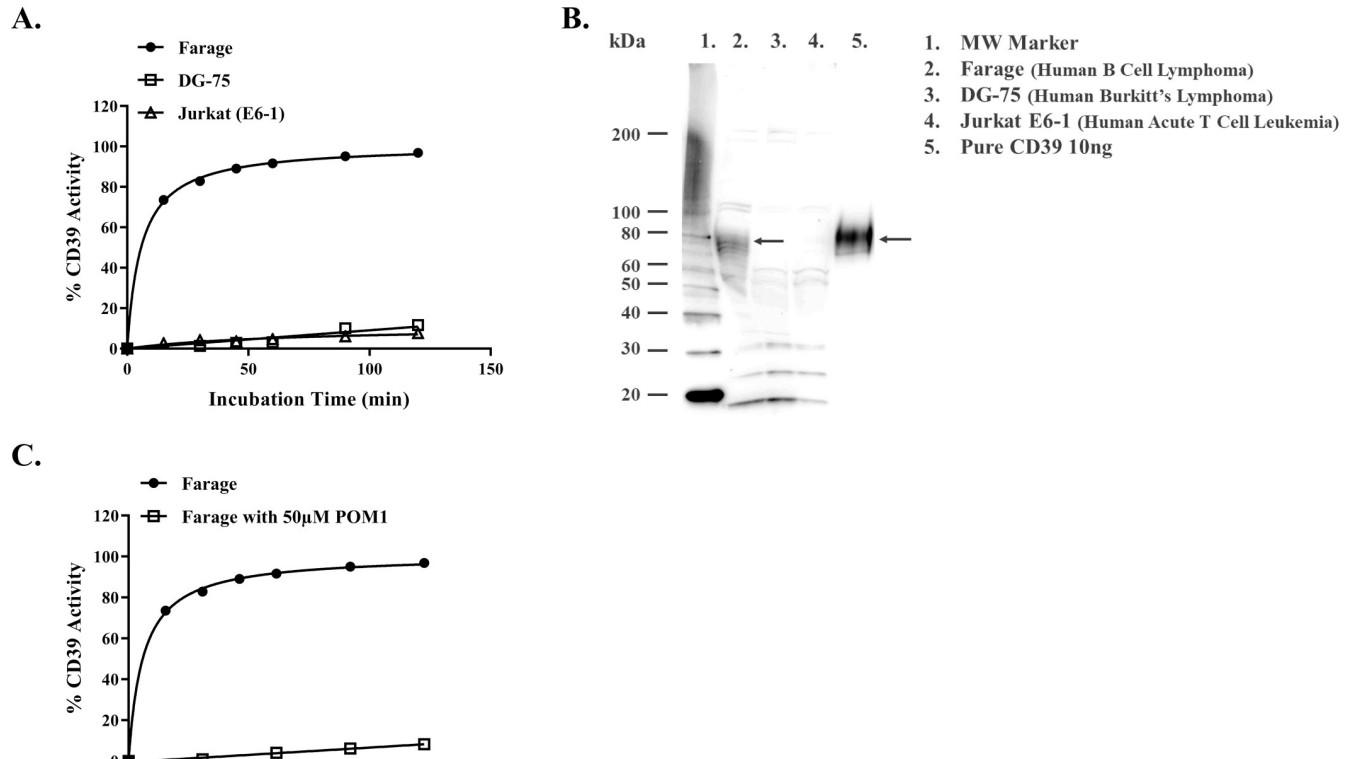

**Fig 9. Determination of membrane bound CD39 enzyme activity and CD39 inhibitor potency on membrane associated CD39 activity.** (A) Determination of CD39 activity in cell membranes using three different cell lines, Farage cell (solid circle) has the highest CD39 activity using ATP as substrate. However, DG-75 (square) and Jurkat-E6-1 (triangle) show low or no CD39 activity. Reactions contained 200k cells per ml with 5μM ATP as substrate. (B) Typical western blotting results of CD39 abundance in membranes of the three cell lines tested in (A). Cell membrane from each cell (20μg per sample) and pure CD39 (10ng) as positive control were used and anti-CD39 antibodies. The experiment of CD39 activity was done in triplicates; results shown are mean ± SD. (c) Monitoring CD39 activity using Farage cells with (square) and without (solid circle) 50μM POM 1. Reactions contained 200K cell per ml of Farage cells per sample and incubated with 5μM ATP at 37˚C with constant shaking at 100rpm. Aliquots of 30μl were withdrawn from each sample at certain time followed by the addition of equal volume of Kinase-Glo® One Solution to monitoring ATP consumption by Farage cells. Each point represents average of a typical experiment done in triplicates; results shown are mean ± SD.

## Conclusion

Current technologies for monitoring the activities of CD39 and CD73 enzymatic activities are based on radioactive substrates such as $^{32}$P-ATP, $^{32}$P-ADP, and $^{32}$P-AMP; or HPLC using UV Absorption detection system of the various nucleotides, or measuring inorganic phosphate using colorimetric detection system. The radioactivity-based method provides accurate results but is health hazardous and generates large amount of radioactive waste. The use of HPLC to monitor the release of products such as ADP or adenosine provides accurate data but requires sophisticated equipment, personnel training, and is not easily amenable to HTS, and most certainly not homogenous. The release of inorganic phosphate as a product of these enzymatic reactions (CD39 and CD73) using ATP and AMP as substrates for CD39 and CD73, respectively, was also used. However, this method is not homogenous, and since it is colorimetric, it does not provide the sensitivity required for low enzyme activity, and encounter interference from phosphate intolerant chemicals. Thus, we believe the bioluminescent platform described here provides, easily to perform, homogenous, sensitive and most important is amenable to

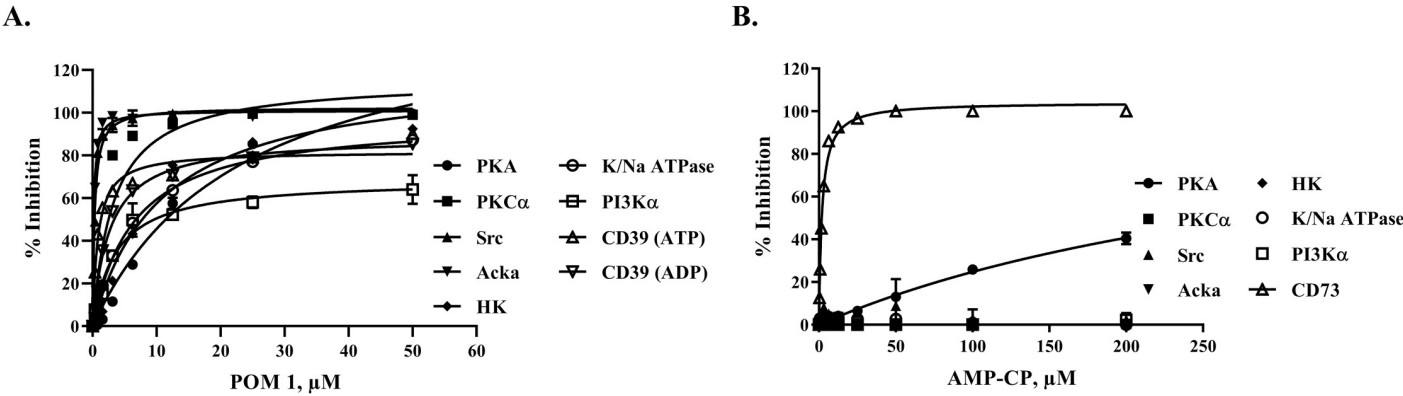

**Fig 10. Inhibitor selectivity using purified and membrane associated CD39.** Inhibitors of CD39 and CD73 were incubated with purified CD39 (A, B) and Farage associated enzyme (C, D) and assayed for CD39 activity using ATP (A, C) or ADP (B, D) as substrate. Results show percentage inhibition of CD39 activity using ARL 67156 (circle), POM1 (solid square), and AMP-CP (triangle). Purified CD39 (0.1ng) and 10μM ATP or ADP in each reaction for 30min at 37˚C; for cell-based Farage cells (200K cells) were incubated with 10μM ATP or ADP in each reaction for 20min at 37˚C. Experiments were carried out in triplicates; results shown are mean ± SD.

**Fig 11. Determining the specificity of CD39 and CD73 inhibitors against other enzymes.** Testing the specificity of CD39 inhibitor POM1 (A) and CD73 inhibitor AMP-CP (B) on the activity of different kinases, ATPase, as well as CD39 and CD73. Enzyme tested are protein serine/threonine/tyrosine, lipid, sugar, and inorganic kinases, and K/Na ATPase. Results show that the majority of enzymes are inhibited by POM1 (A), but the more selective CD73 inhibitor AMP-PC) was highly selective for CD73 with minimal inhibition for PKA and no inhibition of the other enzymes. PKA, protein kinase A (solid circle), PKCα (solid square), Src, protein tyrosine Src kinase (solid triangle), Acka, acetate kinase from *Escherichia coli* (solid reverse triangle), HK, hexokinase from *Saccharomyces cerevisiae* (solid circle), PI3αKinase, p110α/p85α (square), K/K ATPase, Adenosine 5´-Triphosphatase from porcine cerebral cortex (circle), CD39 (ATP substrate, triangle), and CD39 (ADP substrate, reverse triangle); for CD73 (triangle) in panel B. Each point represents average of a typical experiment done in triplicates; results shown are mean ± SD.

HTS which is required for development of novel therapeutics. The assay platform enables monitoring the activity of CD73 and CD39 in their soluble as well as membrane-associated forms. The assays proved to be very sensitive to very low concentration of these enzymes and enable screening for the selectivity of inhibitors of these enzymes in a homogenous format that suits HTS experimental design. The assay platform is the only one that combines determination of both soluble and membrane associated activities of CD73 and CD39. The assays can discriminate between selective inhibitors and promiscuous ones in pure enzyme as well membrane associated enzymes reactions. What is unique about this assay platform is that not only can it detect the effect of small molecule modulators but also of antibodies specific in blocking enzyme activities of these enzymes using intact cells, and thus it can be used to screen for any modulator of these enzymes in purified enzyme as well as membrane associated form.

## Supporting information

**S1 Fig. Schematic representation for detection of CD 39 and CD73 enzyme activities using adenine nucleotides modifying enzymes.** (A) Monitoring the enzyme activity of CD39 using either ATP or ADP as substrate. The principle of the assay is based on the consumption of ATP as substrate by CD39 which can be monitored by determining the amount of ATP remaining in the reaction by an ATP utilizing luciferase reaction. Alternatively, when ADP is used as a substrate, remaining ADP after CD39 reaction can be converted to ATP using adenylate kinase and the ATP generated is determined by an ATP utilizing luciferase reaction. (B) Monitoring the activity of CD73 using AMP substrate and converting remaining AMP in the reaction to ATP via two enzymes (AMP-polyphosphate phosphotransferase and adenylate kinase) and the generated ATP is detected using luciferase reaction.
(TIF)

**S1 Table. List of Antibodies tested including immunogen/Host information.**
(DOCX)

## Author Contributions

**Conceptualization:** Said A. Goueli.

**Data curation:** Said A. Goueli, Kevin Hsiao.

**Formal analysis:** Said A. Goueli, Kevin Hsiao.

**Investigation:** Said A. Goueli.

**Methodology:** Said A. Goueli, Kevin Hsiao.

**Project administration:** Said A. Goueli.

**Supervision:** Said A. Goueli.

**Validation:** Said A. Goueli.

**Visualization:** Said A. Goueli, Kevin Hsiao.

**Writing – original draft:** Said A. Goueli.

**Writing – review & editing:** Said A. Goueli, Kevin Hsiao.

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
