## [Decision Letter · Decision Letter 0]

7 Aug 2019

PONE-D-19-18985

Monitoring and characterizing soluble and membrane-bound ectonucleotidases CD73 and CD39

PLOS ONE

Dear Dr. Goueli,

Thank you for submitting your manuscript to PLOS ONE. After careful consideration, we feel that it has merit but does not fully meet PLOS ONE’s publication criteria as it currently stands. Therefore, we invite you to submit a revised version of the manuscript that addresses the points raised during the review process.

ACADEMIC EDITOR: 

Please refer to reviwers' comments below with particular reference to including increased emphasis on how this approach is superior to currently established protocols e.g. potential to use with live cells. Both reviewers have also commented on the length and number of figures. Please try to more succinctly present the data.

As suggested by reviewer 1, an introduction that includes a wider characterisation of the role of CD73 and CD39 in human physiology/ pathology is warranted.

We would appreciate receiving your revised manuscript by Sep 21 2019 11:59PM. To enhance the reproducibility of your results, we recommend that if applicable you deposit your laboratory protocols in protocols.io, where a protocol can be assigned its own identifier (DOI) such that it can be cited independently in the future. For instructions see: http://journals.plos.org/plosone/s/submission-guidelines#loc-laboratory-protocols

We look forward to receiving your revised manuscript.

Kind regards,

Paul A Beavis

Academic Editor

PLOS ONE

Journal Requirements:

2. Please note that PLOS does not permit references to “data not shown.” Authors should provide the relevant data within the manuscript, the Supporting Information files, or in a public repository. If the data are not a core part of the research study being presented, we ask that authors remove any references to these data.

The authors acknowledge the financial support by Promega Corp for carrying out this research.

We note that you received funding from a commercial source: Promega Corp

Reviewers' comments:

Reviewer's Responses to Questions

**Comments to the Author**

1. Is the manuscript technically sound, and do the data support the conclusions?

Reviewer #1: Yes

Reviewer #2: Partly

2. Has the statistical analysis been performed appropriately and rigorously? 

Reviewer #1: No

Reviewer #2: N/A

3. Have the authors made all data underlying the findings in their manuscript fully available?

Reviewer #1: Yes

Reviewer #2: Yes

4. Is the manuscript presented in an intelligible fashion and written in standard English?

Reviewer #1: Yes

Reviewer #2: No

5. Review Comments to the Author

Reviewer #1: Reviewer comments

Due to the increasing interest in targeting the CD73/CD39 and adenosine receptor axis in the context of immunotherapy to treat cancer, and promising outcomes both in pre-clinical and clinical settings, there is a need to develop new and effective therapeutic agents targeting these receptors. The authors have developed and clearly demonstrated a sensitive assay platform utilizing Promega’s own AMP-GLO or ADP-GLO kits to test the efficacy and specificity of small molecule inhibitors and antibodies for inhibiting the activity of CD39 and CD73. There are however several issues that need to be addressed as outlined below.

Major

A potential limitation of using a luciferase-based detection assay is inhibition of luciferase by inhibitors used. POM1 was mentioned in prior literature to inhibit luciferase in a different enzyme activity kit (doi: 10.4049/jimmunol.1003884). Have the authors confirmed their assay using orthogonal methods, such as alternative enzyme assay or testing their inhibitors at varying concentrations in the absence of CD73 or CD39? It may be important for future screening to include in the protocol a test for potential inhibitors interfering with the output of the AMP-GLO or ADP-GLO kits.

Protein concentration used in western blots for CD79 or CD39 in cells should be shown using either a housekeeping control, or the total protein concentration shown if other methods were used. Alternatively, a FACS based assay could be performed to show expression of cell surface CD79 or CD39.

One of the advantages of this assay as described by the authors is that a homologous detection protocol could be adapted to broad range of kinases and proteins that metabolise ATP. This is great for identifying cross reactivity, which is an advantage over other assays. However, it is difficult to decipher between groups in Figure 17A and 17B. It does appear that at high concentrations of inhibitor POM1, there is inhibition of enzyme activity in each group, but POM1 could be specific at lower doses. Showing the X-axis as a Log10[inhibitor] will make these figures easier to interpret.

Include statistics and p-values.

State competing interests, such as funding from Promega.

Minor

The large number of figures is unnecessary and some streamlining of the figures can make the paper easier to read. Joining Fig 3+4+5, figure 6+7, Figures 8+9+10, figures 12+13, figures 14+15+16 into their respective figures for example. The Figure legends will need to be modified in each case.

Error bars in several figures are hidden behind the point. It may be relevant to show these in front for consistency.

Higher resolution figures should also be provided for most of the figures, this is especially relevant for figure 11.

Including several examples from literature on the advantages of using membrane bound proteins rather than purified proteins in screening for inhibitors, could enhance the rationale of using this approach on live cells. Is it possible to use this assay for primary mouse or human cells, and if so, could this lead to other applications for this assay, such as regulation of enzyme function on cells in response to stimuli, or confirming active inhibition of receptors in vivo after therapy?

Optimization of cell numbers for adherent or non-adherent cells in the assay may be of interest to readers and could be provided as supplementary.

Sufficient references used, covering several studies targeting the CD39/CD73 and adenosine axis in combination with checkpoint inhibitors to treat tumors. Identified its importance in tumor escape, regulation of anti-tumor immunity (both myeloid and lymphoid populations) in both preclinical and clinical studies. Increased tumor CD73 and CD39 expression correlate with poorer patient outcomes. It may be prudent to include J.Stagg 2010 PNAS paper, which is the original paper that utilized anti CD73 against cancer. More recent papers have also linked CD39 expression as a marker of tumor-reactive T cells in tumor infiltrating CD8 populations in both primary and metastatic tumors. Citing these in the text can be relevant.

Relating to the methods for preparation of adherent cells for the assay, it was not clearly mentioned if these cells need to be re-trypsinized after culture overnight in clear bottom plate to be transferred into 96 well plate?

Few spelling errors throughout the paper to be corrected.

Reviewer #2: The manuscript highlights a system to accurately assess the activity of CD73 and CD39 in vitro, an important assay as CD73 and CD39 inhibitors and antibodies are of increasing interest clinically. However, the manuscript requires a number of changes to improve and increase its readability for a greater audience. Many of the sections could be condensed to highlight further the main message of the manuscript succinctly.

Major points:

The introduction is long and highlights much detail about the combinatorial potential of targeting the adenosinergic pathway alongside other therapeutic modalities in cancer, the authors should consider whether their may be a more appropriate introduction to provide a better understanding to their results and the assay they have developed.

Methods section provides numerous details about the reagents used, however, this could be more strategically integrated within the methods.

Please highlight the clones of antibodies being used in addition to their catalogue number. Ab71322 is no longer available from the company and the authors should reconsider use of this data.

In scheme 1 – please include in part (a) adenylate kinase as mentioned in the legend for this section. In (b) please keep the names of the reagents or symbols used to identify them consistent to that in (a).

The number of figures is excessive, many could be combined to make a multiple figure panel, increasing the flow of the paper. Within the figure legends it describes that these figures are performed in triplicate, is that biological or technical triplicates?

In Figure 1 – the inset graph in panel A is not well defined.

Figure 2 – the legend references schematic 1A, it is unclear how this section of the schema relates to the figure, should this be a reference to Scheme 1B where CD73 activity is performed? Figure 2 is also not referenced by number in the body of the text (see line 415 “as shown in figure”).

Figure 3 – uses the same concentration of AMP, in line 433 in reference to this figure the text states “…using different concentrations of AMP”. Please keep units consistent within the manuscript ie in this figure the units used are ng whereas pg has been used in the text. Similarly, line 466 includes mU which are not utilized elsewhere.

Figure 11 – please highlight which antibody number relates to which antibody and the specific clone (this may be most practical to do in the methods section). Some antibodies within the methods section highlight the same clone from different companies being used in this assay, please detail their reproducibility. In addition, many of the CD73 antibodies did not inhibit with CD73 enzymatic activity, please discuss whether this is in keeping with what is known about that antibodies specificity. Line 543 preceding this figure also states that “control antibodies showed minimal effect…” it is unclear within the figure which control antibodies are used and whether any in addition to CD39 are apparent.

Figure 13 – Anti-CD39 was also used in this figure as a negative control for CD73 activity – it would be of interest to look at this antibodies ability to impede CD39 enzymatic activity alongside POM1 and ARL67156.

In Figure 17, POM1 is shown to lack specificity to CD39 and instead inhibits multiple enzymes, it would be of value to comment on the use of POM1 to delineate CD39-mediated anti-tumor efficacy.

In general, the figure legends contain a large amount of information that belongs in the material and methods, please look to make these more succinct.

It would be of interest to utilize this system to measure CD73 and CD39 activity, or the inhibition of enzymatic activity from samples ex vivo rather than in vitro. Can this system be used to measure the presence and activity of CD73 and CD39 inhibitors in circulation? The presence of increased soluble CD73/CD39 or exosome-derived CD73/CD39? Or the enzymatic activity in the tumor microenvironment ex vivo? Please comment or display the utility of this assay in these settings.

Please comment on the advantages/disadvantages of this system compared to those currently utilized. In addition, very limited discussion of the impact of the results is provided throughout and a greater emphasis should be placed towards ensuring this is clear to the reader.

Minor points:

Please check the manuscript for consistency regarding CD73, CD39 and the Farage cell line as well as other nomenclature.

Please provide some reference to the differences in temperature used within the assays. For example, figure 2 at 23 degrees C while Figure 12 is 37 degrees.

Please see some small errors to correct:

Line 19 – increased to increase

Line 20 /65– immunosuppressant, generally refers to a drug type reconsider use of this word throughout.

Line 22/25 – CD39 while upstream of CD73 and the conversion of AMP to adenosine, should not be referres to as a major source for adenosine.

Line 34 – specify the what novel inhibitors you refer too for cancer therapy.

Line 36 – simplify “immune-checkpoint blockers drugs”

Line 70/71 – Rephrase sentence

Line 92 – Replace effector with CD8+ T cells

Line 104/105 – Please explain low cross reactivity with other nucleotides? Do you mean ectonucleotidases?

Line 115 – Fc should have the gamma symbol not that currently shown.

Line 116/118 – refer as A2AR inhibitor or antagonist, not anti A2AR.

Line 121 – AZD4635’s remove the ‘s

Line 124/126 – rephrase sentence

Line 133 – why refer to them as tentative therapeutic approaches?

Line 136 – should read “as well as the tumor itself”

Line 137 – should read monitors

Line 157 – EMEM?

Line 166 – rephrase kinases kits.

Line 272 – per what well?

Line 284 – A.1.?

Line 354 – should read “a very potent immunosuppressive”

Line 354 – 365 – redundant or better included in the introduction please revise.

Line 369/370 – relevance for the discussion of the bacterial enzyme within this paper?

Line 425 – delete i.e.,

Line 444 – should read “Fig. 4, CD73..”

Line 534 – Delete three.

Line 540/541 – refers to CD39 antibodies should read antibody as only one is used and the use of control antibodies which are not clearly shown within the figure should be clarified.

Line 564 – should read “while the other monitors”

Line 608 – remove the use of totally.

Line 656 – should read panel not penal.

Figure 13/17 legends please check for incorrectly formatted symbols.

Line 659 replace that with to.

6. PLOS authors have the option to publish the peer review history of their article (what does this mean?). If published, this will include your full peer review and any attached files.

Reviewer #1: No

Reviewer #2: No

---

## [Author Response · Author response to Decision Letter 0]

30 Aug 2019

Dear Reviewers:

Your comments are shown in BLACK text and our response to your questions are provided in RED

5. Review Comments to the Author

Reviewer #1: Reviewer comments

Due to the increasing interest in targeting the CD73/CD39 and adenosine receptor axis in the context of immunotherapy to treat cancer, and promising outcomes both in pre-clinical and clinical settings, there is a need to develop new and effective therapeutic agents targeting these receptors. The authors have developed and clearly demonstrated a sensitive assay platform utilizing Promega’s own AMP-GLO or ADP-GLO kits to test the efficacy and specificity of small molecule inhibitors and antibodies for inhibiting the activity of CD39 and CD73. There are however several issues that need to be addressed as outlined below.

Major

A potential limitation of using a luciferase-based detection assay is inhibition of luciferase by inhibitors used. POM1 was mentioned in prior literature to inhibit luciferase in a different enzyme activity kit (doi: 10.4049/jimmunol.1003884). Have the authors confirmed their assay using orthogonal methods, such as alternative enzyme assay or testing their inhibitors at varying concentrations in the absence of CD73 or CD39? It may be important for future screening to include in the protocol a test for potential inhibitors interfering with the output of the AMP-GLO or ADP-GLO kits.

We have tested POM1 on the luciferase activity as control and we found no inhibitory effect of the compound on its activity. Although we have not tested this assay using orthogonal assay, very recent publication used this technology (AMP Glo) to develop what they called “an exceptionally potent inhibitor of human CD37” (Bowman CE, da Silva RG, Pham A, Young SW. An exceptionally potent inhibitor of human CD73. Biochemistry 2019; 58: 3331-3334) and compared their kinetic data using inorganic phosphate release and the data confirmed the validity of bioluminescence approach. Furthermore, since we launched ADP Glo for kinases and ATPases, it has been cited in over 1500 citations as well as numerus patents, and it is the main assay used for large scale screening for inhibitors in HTS laboratories. Besides, we have not found potent inhibitors of this assay other than few luciferase inhibitors which would be excluded when controls were run simultaneously. 

Protein concentration used in western blots for CD39 or CD39 in cells should be shown using either a housekeeping control, or the total protein concentration shown if other methods were used. Alternatively, a FACS based assay could be performed to show expression of cell surface CD79 or CD39.

We have provided the protein concentrations used in the legends of the two figures (Fig 5 for CD73 and Figure 9 for CD39). We have used 10 micrograms of cell lysate protein in Figure 5 and 20 micrograms of cell lysate protein in Figure 9. 

One of the advantages of this assay as described by the authors is that a homologous detection protocol could be adapted to broad range of kinases and proteins that metabolize ATP. This is great for identifying cross reactivity, which is an advantage over other assays. However, it is difficult to decipher between groups in Figure 17A and 17B. It does appear that at high concentrations of inhibitor POM1, there is inhibition of enzyme activity in each group, but POM1 could be specific at lower doses. Showing the X-axis as a Log10[inhibitor] will make these figures easier to interpret.

Include statistics and p-values.

State competing interests, such as funding from Promega.

We have replotted Figure 17 A and B (Now Fig 11 A and B) to highlight the lower concentrations of POM1 tested against other kinases and as clearly shown that the compound inhibits these kinases at low concentrations which is not surprising since it inhibits ATPase activity of CD39 and thus it appears to be a competitive inhibitor towards ATP as substrate. It is noteworthy that POM1, up to 50 Micromolar, does not inhibit CD73 which uses AMP as substrate as shown in Figure 10. 

We also indicated that Promega supported this research. 

Minor

The large number of figures is unnecessary and some streamlining of the figures can make the paper easier to read. Joining Fig 3+4+5, figure 6+7, Figures 8+9+10, figures 12+13, figures 14+15+16 into their respective figures for example. The Figure legends will need to be modified in each case.

Error bars in several figures are hidden behind the point. It may be relevant to show these in front for consistency.

Higher resolution figures should also be provided for most of the figures, this is especially relevant for figure 11.

We have implemented the reviewer’s recommendations by combining the figures and now we have 11 figures plus the scheme instead of 17 figures plus the scheme. Error bars are clearly shown. Also, we have increased the resolutions of those figures as recommended.

Including several examples from literature on the advantages of using membrane bound proteins rather than purified proteins in screening for inhibitors, could enhance the rationale of using this approach on live cells. Is it possible to use this assay for primary mouse or human cells, and if so, could this lead to other applications for this assay, such as regulation of enzyme function on cells in response to stimuli, or confirming active inhibition of receptors in vivo after therapy?

We do not anticipate encountering problems using this assay for other cell types such as primary mouse or human cells since it was tested with many different cell types as long as these two enzymes CD39 and CD73 are expressed on the surface of cell membrane and thus their catalytic activity can be monitored using substrates added to the medium. 

Optimization of cell numbers for adherent or non-adherent cells in the assay may be of interest to readers and could be provided as supplementary. 

We have described how we optimized cell number for detection of enzyme activities

Sufficient references used, covering several studies targeting the CD39/CD73 and adenosine axis in combination with checkpoint inhibitors to treat tumors. Identified its importance in tumor escape, regulation of anti-tumor immunity (both myeloid and lymphoid populations) in both preclinical and clinical studies. Increased tumor CD73 and CD39 expression correlate with poorer patient outcomes. It may be prudent to include J. Stagg 2010 PNAS paper, which is the original paper that utilized anti CD73 against cancer. More recent papers have also linked CD39 expression as a marker of tumor-reactive T cells in tumor infiltrating CD8 populations in both primary and metastatic tumors. Citing these in the text can be relevant.

We are aware of the contribution of Dr. Stagg to the field , and in fact we have already cited him in references #18 and 19. Although we have included several references on the role of CD39 in tumor responses, we also added the reference Duhen, et al (Ref#22) as recommended by the reviewer which focused on CD39 role in CD8 T Cells in solid tumors. 

18 Allard B, Pommey S, Smyth MJ, Stagg J. Targeting CD73 enhances the antitumor activity of anti-PD-1 and anti-CTLA-4 mAbs. Clinical Cancer Res, 2013; 19: 5626-5635

19 Beavis, P.A, Stagg, J, Darcy, P.K, Smyth, M.J. CD73: a potent suppressor of antitumor immune responses Trends in Immunology 2012; 33: 231-237

20 Xu S, Shao QQ, Sun JT, Yang N, Xie Q, Wang DH et al. Synergy between the ectoenzymes CD39 and CD73 contributes to adenosinergic immunosuppression in human malignant gliomas. Neuro Oncology 2013;15: 1160-1172

22. Duhen T, Duhen R, Montler R, Moses J, Moudgil T, de Miranda NF, et al. Co-expression of CD39 and CD103 identifies tumor-reactive CD8 T cells in human solid tumors. Nature Commun.2018; 9:1-13

Relating to the methods for preparation of adherent cells for the assay, it was not clearly mentioned if these cells need to be re-trypsinized after culture overnight in clear bottom plate to be transferred into 96 well plate?

We did not need to re-trypsinize cells because cells were trypsinized and suspended for counting before aliquoting into wells and left overnight to attach and ready for treatment. 

Few spelling errors throughout the paper to be corrected.

These have been taken care as advised

Reviewer #2: The manuscript highlights a system to accurately assess the activity of CD73 and CD39 in vitro, an important assay as CD73 and CD39 inhibitors and antibodies are of increasing interest clinically. However, the manuscript requires a number of changes to improve and increase its readability for a greater audience. Many of the sections could be condensed to highlight further the main message of the manuscript succinctly.

Major points:

The introduction is long and highlights much detail about the combinatorial potential of targeting the adenosinergic pathway alongside other therapeutic modalities in cancer, the authors should consider whether there may be a more appropriate introduction to provide a better understanding to their results and the assay they have developed.

It is our understanding that the main interest in monitoring the activities of CD39 and CD73 is that they ultimately generate adenosine and thus they are highlighted as drug targets since adenosine and adinosinergic modulators are intimately involved in immunosuppression and improving the microenvironment for the tumor. We have emphasized the role of these two enzymes as drug targets with the inclusion of biochemical; small molecule inhibitors as well as monoclonal antibodies targets these enzymes as potential therapeutics. In fact, most of the current literature show a strong interest and an increased clinical trial targeting these enzymes. We believe the introduction embodies the two targeted enzymes, the therapeutic potential, and their clinical relevance as targeted by small and large molecules modulators. It is noteworthy that one reviewer wanted to expand it to include additional information which we briefly did and added one additional reference

Methods section provides numerous details about the reagents used; however, this could be more strategically integrated within the methods.

We have condensed part of the method section as recommended. 

Please highlight the clones of antibodies being used in addition to their catalogue number. Ab71322 is no longer available from the company and the authors should reconsider use of this data.

We understand that the clone Ab71322 may no longer available, but when we did these studies, we were able to get it for our studies. Other antibodies are continually being made and future manuscripts might expand the list as long as clinical interest continue for these targets.

In scheme 1 – please include in part (a) adenylate kinase as mentioned in the legend for this section. In (b) please keep the names of the reagents or symbols used to identify them consistent to that in (a).

Since adenylate kinase is a component of AMP-Glo Reagent II, we added it within the label AMP-Glo Reagent II 

The number of figures is excessive, many could be combined to make a multiple figure panel, increasing the flow of the paper. Within the figure legends it describes that these figures are performed in triplicate, is that biological or technical triplicates?

We have followed the advice of the reviewers and combined the figures as was recommended. Figures were condensed into 11 instead of 17

These experiments were carried out three times and the results show the mean +/- SD

In Figure 1 – the inset graph in panel A is not well defined.

The inset is an expanded scale of AMP concentrations shown in the main figure to show that the AMP can be detected at very low concentrations. 

Figure 2 – the legend references schematic 1A, it is unclear how this section of the schema relates to the figure, should this be a reference to Scheme 1B where CD73 activity is performed? Figure 2 is also not referenced by number in the body of the text (see line 415 “as shown in figure”).

Figure 2 was listed on line 422 of the original manuscript. We followed the reviewer recommendation and made the change

Figure 3 – uses the same concentration of AMP, in line 433 in reference to this figure the text states “…using different concentrations of AMP”. Please keep units consistent within the manuscript ie in this figure the units used are ng whereas pg has been used in the text. Similarly, line 466 includes mU which are not utilized elsewhere.

Regarding AMP concentrations, we corrected the text on line 433 of the original manuscript. The reason we use ng and pg when necessary, is to avoid the inclusion of many decimals, and thus, whole numbers are easy to read as long as we specify ng or pg. We understand that for cN-II, units were used since we did not receive protein concentration from the vendor and thus since we are using relative activity, the lower the units or amount of enzyme the more sensitive the assay. Also, percent inhibition will not be affected by using enzyme units or protein concentration.

Figure 11 – please highlight which antibody number relates to which antibody and the specific clone (this may be most practical to do in the methods section). Some antibodies within the methods section highlight the same clone from different companies being used in this assay, please detail their reproducibility. In addition, many of the CD73 antibodies did not inhibit with CD73 enzymatic activity, please discuss whether this is in keeping with what is known about that antibody’s specificity. Line 543 preceding this figure also states that “control antibodies showed minimal effect…” it is unclear within the figure which control antibodies are used and whether any in addition to CD39 are apparent.

These antibodies all recognize CD73 and they were ordered from the vendors as CD73 antibodies with only one that is made against CD39. Of these antibodies, two recognize the same epitope on CD73 but from two different vendors to test the reproducibility of our assay against membrane bound CD73. We have no idea in advance of our work whether these antibodies inhibit CD73 enzymatic activity since the data sheets we received from the vendors show only western blots and purity of the antibodies. I believe we are the first to show the reactivity of these antibodies against CD73 enzymatic activity and thus we think these are valuable information for the field. It appears that some of these antibodies recognize different motifs on CD73 which does not affect its enzymatic activity. The two controls in our studies are CD39 selective antibody and no antibody addition. 

Figure 13 – Anti-CD39 was also used in this figure as a negative control for CD73 activity – it would be of interest to look at this antibodies ability to impede CD39 enzymatic activity alongside POM1 and ARL67156.

We have not carried out this particular experiment, but we anticipate combined effect) additive, synergistic, or no effect if they bind to the same catalytic site. Since we are monitoring the activity, it appears that inhibition would be predicted but might be synergistic, i.e., lower concentration of both small molecule and antibody might lower the concentration of either one alone.

In Figure 17, POM1 is shown to lack specificity to CD39 and instead inhibits multiple enzymes, it would be of value to comment on the use of POM1 to delineate CD39-mediated anti-tumor efficacy.

I believe we have highlighted this point in our discussion as indicated that due to the lack of specificity of this inhibitor, and unless a very selective inhibitor is found for CD39, we recommend targeting CD73 if small molecules will be tested since the small molecule inhibitor of CD73 shows high selectivity at low concentration. However, due to the high selectivity of monoclonal antibodies towards their targets, it might be appropriate to target either one or both targets. 

In general, the figure legends contain a large amount of information that belongs in the material and methods, please look to make these more succinct.

We have condensed the figure into 11 instead of 17 in the previous manuscript and thus the legends were also combined and made succinct.

It would be of interest to utilize this system to measure CD73 and CD39 activity, or the inhibition of enzymatic activity from samples ex vivo rather than in vitro. Can this system be used to measure the presence and activity of CD73 and CD39 inhibitors in circulation? The presence of increased soluble CD73/CD39 or exosome-derived CD73/CD39? Or the enzymatic activity in the tumor microenvironment ex vivo? Please comment or display the utility of this assay in these settings.

We do not anticipate encountering problems using this assay for other cell types such as primary mouse or human cells since it was tested with many different cells as long as these two enzymes CD39 and CD73 are expressed on the surface of cell membrane and thus their catalytic activity can be monitored using substrates added to the medium. Similarly, the activity of these two enzymes can be also done with exosome derived CD39/CD73. Regarding the presence of CD39 and CD73 in circulations, it would be advisable to immunoprecipitate these enzymes using appropriate antibodies for enrichments and increased detectability. 

Please comment on the advantages/disadvantages of this system compared to those currently utilized. In addition, very limited discussion of the impact of the results is provided throughout and a greater emphasis should be placed towards ensuring this is clear to the reader.

Current technologies for monitoring the activities of these enzymes are based on radioactive substrates such as 32P-ATP, 32P-ADP, and 32P-AMP which is health hazardous and generates large amount of radioactive waste. Other methods rely on the use of HPLC to monitor the release of products such as ADP or adenosine. This approach provides accurate data but requires sophisticated equipment, personnel training, and not easily amenable to HTS, and most certainly not homogenous. It is noteworthy that bioluminescence data we generated showed its higher sensitivity than HPLC at the nanomolar concentrations. One other approach is using detection of released inorganic phosphate from both ATP and AMP as substrates for both enzymes respectively. However, this method is not homogenous, not sensitive enough, and encounter interference from phosphate intolerant chemicals. Thus, we believe this platform described here provides, easily to perform, homogenous, sensitive and above amenable to HTS which is required for development of novel therapeutics. 

Minor points:

Please check the manuscript for consistency regarding CD73, CD39 and the Farage cell line as well as other nomenclature.

Please provide some reference to the differences in temperature used within the assays. For example, figure 2 at 23 degrees C while Figure 12 is 37 degrees.

The activity assays were carried out at the optimal temperatures provided by the manufacturer for these enzymes and therefore, we followed their enzymatic assay conditions.

Please see some small errors to correct:

Line 19 – increased to increase

Done

Line 20 /65– immunosuppressant, generally refers to a drug type reconsider use of this word throughout.

Done

Line 22/25 – CD39 while upstream of CD73 and the conversion of AMP to adenosine, should not be referred to as a major source for adenosine.

We respectfully referred to the combined effect of CD39 and CD73 as the major source for adenosine in the tumor microenvironment 

Line 34 – specify the what novel inhibitors you refer too for cancer therapy.

The sentence was corrected to may lead to more effective cancer therapy and we included the following two references. Junker, A et al (2019) Structure relationship of purine and pyrimidine nucleotides as ect-5’-nucleotidase (CD73) inhibitors. J Med. Chem 62: 3677-3695) and Bowman, CE et al (2019) An exceptionally potent inhibitor of human CD73. Biochemistry 58: 3331-3334

Line 36 – simplify “immune-checkpoint blockers drugs”

Simplified to immune checkpoint inhibitors 

Line 70/71 – Rephrase sentence

The sentence was divided into two for readability 

Line 92 – Replace effector with CD8+ T cells

Done

Line 104/105 – Please explain low cross reactivity with other nucleotides? Do you mean ectonucleotidases?

Because this antibody does not bind to the AMP binding site, it indicates that AMP binding is not involved in the inhibition and consequently any other nucleotides will not be relevant in inhibiting CD73.

Line 115 – Fc should have the gamma symbol not that currently shown.

Corrected

Line 116/118 – refer as A2AR inhibitor or antagonist, not anti A2AR.

Corrected

Line 121 – AZD4635’s remove the ‘s

Done

Line 124/126 – rephrase sentence

Done

Line 133 – why refer to them as tentative therapeutic approaches?

We changed it to potential therapeutic approaches

Line 136 – should read “as well as the tumor itself”

Fixed

Line 137 – should read monitors

Done

Line 157 – EMEM?

Eagle's Minimum Essential Medium, Catalog No. 30-2003 from ATCC.

Line 166 – rephrase kinases kits. 

(done)

Line 272 – per what well?

Response rewritten

Line 284 – A.1.?

Corrected

Line 354 – should read “a very potent immunosuppressive”

Reworded

Line 354 – 365 – redundant or better included in the introduction please revise.

Line 369/370 – relevance for the discussion of the bacterial enzyme within this paper?

Thus, not only the phylogenetic classification of this enzyme during evolution (bacterial and eukaryotic) is important, but also within the eukaryotes, the cellular localization of the enzyme lead to diverse substrate preference for these isoforms

Line 425 – delete i.e.,

Done

Line 444 – should read “Fig. 4, CD73.”

Corrected

Line 534 – Delete three.

Removed

Line 540/541 – refers to CD39 antibodies should read antibody as only one is used and the use of control antibodies which are not clearly shown within the figure should be clarified.

Single antibody was used, and the wording was corrected

Line 564 – should read “while the other monitors”

Done

Line 608 – remove the use of totally.

Removed

Line 656 – should read panel not penal.

Done

Figure 13/17 legends please check for incorrectly formatted symbols.

Line 659 replace that with to.

Done

---

## [Decision Letter · Decision Letter 1]

25 Sep 2019

PONE-D-19-18985R1

Monitoring and characterizing soluble and membrane-bound ectonucleotidases CD73 and CD39

PLOS ONE

Dear Dr. Goueli,

Thank you for submitting your manuscript to PLOS ONE. After careful consideration, we feel that it has merit but does not fully meet PLOS ONE’s publication criteria as it currently stands. Therefore, we invite you to submit a revised version of the manuscript that addresses the points raised during the review process.

ACADEMIC EDITOR: 

There are some minor points that require a bit more attention to make the manuscript acceptable for publication. Please refer to the comments of reviewer 2 with regards to Figure 7 in particular.

We would appreciate receiving your revised manuscript by Nov 09 2019 11:59PM. To enhance the reproducibility of your results, we recommend that if applicable you deposit your laboratory protocols in protocols.io, where a protocol can be assigned its own identifier (DOI) such that it can be cited independently in the future. For instructions see: http://journals.plos.org/plosone/s/submission-guidelines#loc-laboratory-protocols

We look forward to receiving your revised manuscript.

Kind regards,

Paul A Beavis

Academic Editor

PLOS ONE

Reviewers' comments:

Reviewer's Responses to Questions

**Comments to the Author**

1. If the authors have adequately addressed your comments raised in a previous round of review and you feel that this manuscript is now acceptable for publication, you may indicate that here to bypass the “Comments to the Author” section, enter your conflict of interest statement in the “Confidential to Editor” section, and submit your "Accept" recommendation.

Reviewer #1: All comments have been addressed

Reviewer #2: (No Response)

2. Is the manuscript technically sound, and do the data support the conclusions?

Reviewer #1: Yes

Reviewer #2: Partly

3. Has the statistical analysis been performed appropriately and rigorously? 

Reviewer #1: N/A

Reviewer #2: N/A

4. Have the authors made all data underlying the findings in their manuscript fully available?

Reviewer #1: Yes

Reviewer #2: Yes

5. Is the manuscript presented in an intelligible fashion and written in standard English?

Reviewer #1: Yes

Reviewer #2: Yes

6. Review Comments to the Author

Reviewer #1: It will be prudent to go over the text and legends again before final submission as I have spotted several grammar and spelling errors that haven't been corrected. Figure 1A legend needs clarification, as it is unclear if the figures shown are performed in the presence or absence of 100uM ATP? Aside from these, the paper is much more readable and most comments have been sufficiently addressed.

Reviewer #2: Some points were addressed but there remains a major need to provide appropriate level of detail for the interpretation of figures.

Major points:

- Figure 7 – antibody details must be provided within – it is not sufficient to label with just Ab1, Ab2 etc without detailing which antibody clone it relates to. It is unclear which of the bars represents the control (non-CD73/CD39 antibodies), these are essential details in the interpretation of the graph. In addition, clones for all antibodies should be detailed within methods, not just catalog info.

- The figure quality is very poor and must be improved for publication. Labels are difficult to read as they are in many cases blurry.

Minor points:

Authors should check for spelling and grammatical errors throughout the manuscript see below for some identified.

- Correct the misspelling of Farage cell line, often referred to as Farag as well as Jurkat referred to as Jurkate.

- Define A2AR at the first time it is abbreviated.

- Line 34 – delete (,)

- Line 51 – should read “leading candidate in…”

- Line 57 – Ref 8 is strange – primary article by Bastid, Cancer Immunology Research, 2015 would serve this statement better and broader across multiple cancers.

- Line 70 – delete ‘That…’

- Line 119 – should read checkpoint

- Line 115 and 120 – delete “anti” in relation to A2AR – misleading suggestive that it’s an antibody, please ensure this is removed throughout the text.

- Line 123 – should read A2AR

- Line 142 – delete (s)

- Line 221 – 96-well is repeated

- Line 259 – should read ‘or for a certain time period’

- Line 267 – aberrant )

- Line 272 – should read “known CD39 inhibitor…”

- Line 334 – should read 80-90%

- Line 355 – should read “, 10 min”

- Line 390 – should read “remaining”

- Line 621 – should read “antibody”

- Line 711 – should read from not form.

- Line 757 - should read ‘a diverse set’

- Line 763 - should read ‘that because POM1’

- Line 789 – should read ‘and is not easily’

- Line 805 – should read ‘not only can it’

- Reference 22 and 29 is listed as the same

7. PLOS authors have the option to publish the peer review history of their article (what does this mean?). If published, this will include your full peer review and any attached files.

Reviewer #1: No

Reviewer #2: No

---

## [Author Response · Author response to Decision Letter 1]

2 Oct 2019

Please see the cover letter to the editor and response to the reviewers in which all concerns have been addressed and all questions have been answered as requested.

---

## [Editor Report · Decision Letter 2]

8 Oct 2019

Monitoring and characterizing soluble and membrane-bound ectonucleotidases CD73 and CD39

PONE-D-19-18985R2

Dear Dr. Goueli,

We are pleased to inform you that your manuscript has been judged scientifically suitable for publication and will be formally accepted for publication once it complies with all outstanding technical requirements.

With kind regards,

Paul A Beavis

Academic Editor

PLOS ONE

Additional Editor Comments (optional):

Thank you for making the remaining changes to the manuscript.

Reviewers' comments:

N/A.

---

## [Editor Report · Acceptance letter]

17 Oct 2019

PONE-D-19-18985R2 

Monitoring and characterizing soluble and membrane-bound ectonucleotidases CD73 and CD39 

Dear Dr. Goueli:

I am pleased to inform you that your manuscript has been deemed suitable for publication in PLOS ONE. Congratulations! Your manuscript is now with our production department. 

With kind regards,

on behalf of

Dr. Paul A Beavis 

Academic Editor

PLOS ONE